# On Fenchel Mini-Max Learning

**Chenyang Tao[1], Liqun Chen[1], Shuyang Dai[1], Junya Chen[1,2], Ke Bai[1], Dong Wang[1],**
**Jianfeng Feng[3], Wenlian Lu[2], Georgiy Bobashev[4], Lawrence Carin[1]**

[1] Electrical & Computer Engineering, Duke University, Durham, NC, USA
[2] School of Mathematical Sciences, Fudan University, Shanghai, China
[3] ISTBI, Fudan University, Shanghai, China
[4] RTI International, Research Triangle Park, NC, USA
{chenyang.tao, lcarin}@duke.edu

## Abstract

Inference, estimation, sampling and likelihood evaluation are four primary goals of probabilistic modeling. Practical considerations often force modeling approaches to make compromises between these objectives. We present a novel probabilistic learning framework, called Fenchel Mini-Max Learning (FML), that accommodates all four desiderata in a flexible and scalable manner. Our derivation is rooted in classical maximum likelihood estimation, and it overcomes a longstanding challenge that prevents unbiased estimation of unnormalized statistical models. By reformulating MLE as a mini-max game, FML enjoys an unbiased training objective that ($i$) does not explicitly involve the intractable normalizing constant and ($ii$) is directly amendable to stochastic gradient descent optimization. To demonstrate the utility of the proposed approach, we consider learning unnormalized statistical models, nonparametric density estimation and training generative models, with encouraging empirical results presented.

## 1   Introduction

When learning a probabilistic model, we are typically interested in one or more of the following operations:

- *Inference*: Represent observation $x \in \mathbb{R}^p$ with an informative feature vector $z \in \mathbb{R}^d$, ideally with $d \ll p$; $z$ is often a latent variable in a model of $x$.
- *Estimation*: Given a statistical model $p_\theta(x)$ for data $x$, learn model parameters $\theta$ that best describe the observed (training) data.
- *Sampling*: Efficiently synthesize samples from $p_\theta(x)$ given learned $\theta$, with drawn $x \sim p_\theta(x)$ faithful to the training data.
- *Likelihood evaluation*: With learned $\theta$ for model $p_\theta(x)$, calculate the likelihood of new $x$.

One often makes trade-offs between these goals, as a result of practical considerations (*e.g.*, computational efficiency); see Table S1 in the Supplementary Material (SUPP) for a brief summary. We are particularly interested in the case for which the model $\tilde{p}_\theta(x)$ is unnormalized; *i.e.*, $\int \tilde{p}_\theta(x)dx = Z(\theta) \neq 1$, with $Z(\theta)$ difficult to compute [49].

Maximum likelihood estimation (MLE) is widely employed in the training of probabilistic models [11, 22], in which the expected log-likelihood $\log p_\theta(x)$ is optimized wrt $\theta$, based on the training examples. For unnormalized model density function $\tilde{p}_\theta(x) = \exp(-\psi_\theta(x))$, where $\psi_\theta(x)$ is the potential function and $\theta$ are the model parameters, the likelihood is $p_\theta(x) = \frac{1}{Z(\theta)}\tilde{p}_\theta(x)$. The partition function $Z(\theta)$ is typically not represented in closed-form when considering a flexible choice of $\psi_\theta(x)$, such as a deep neural network. This makes the learning of unnormalized models particularly challenging, as the gradient computation requires an evaluation of the integral. In practice, this integral is approximated with averaging over a finite number of Monte Carlo samples. However, using the existing finite-sample Monte Carlo estimate of $Z_\theta$ will lead to a biased approximation of

the $\log$-likelihood objective (see Section 2.1). This issue is aggravated as the dimensionality of the problem grows.

Many studies have been devoted to addressing the challenge of estimation with unnormalized statistical models. Geyer [23, 24] proposed Markov chain Monte Carlo MLE (MCMC-MLE), which employs a likelihood-ratio trick. Contrastive divergence (CD) [33] directly estimates the gradient by taking MCMC samples. Hyvärinen [36] proposed score matching (SM) to estimate an unnormalized density, bypassing the need to take MCMC samples. Noise contrastive estimation (NCE) learns the parameters for unnormalized statistical models via discriminating empirical data against noise samples [28, 29]. This concept can be further generalized under the Bregman divergence [27]. More recently, dynamic dual embedding (DDE) explored a primal-dual view of MLE [15, 16], while Stein implicit learning (SIL) [46, 41] and kernel score estimation [60] match the landscape of the potential with that of kernel-smoothed empirical observations. However, these approaches are susceptible to poor scalability (SM, MCMC-MLE), biased estimation (CD), and computational (DDE, SIL) and statistical (NCE) efficiency issues.

Concerning design of models that yield realistic drawn samples, considerable recent focus has been placed on *implicit generative models* [48], which include the generative adversarial network (GAN) [25, 51, 4, 61], the generative moment matching network (GMMN) [42, 19], implicit MLE (IMLE) [39], among others. In this setting one typically doesn't have an explicit $\tilde{p}_\theta(x)$ or $p_\theta(x)$, and the goal is to build a model of the data generation process directly. Consequently, such schemes typically have difficulty addressing the aforementioned likelihood goal. Additionally, such models often involve training strategies that are challenging due to instabilities or expressiveness, such as adversarial estimation (GAN) and kernelized formulation (GMMN).

For these reasons, likelihood-based models remain popular. Among them variational inference (VI) [6] and generative flows (FLOW) [56, 53] are two of the most promising directions, and have undergone rapid development recently [66]. Despite this progress, challenges remain. The variational bound employed by VI is often not sufficiently tight in practice (undermining the likelihood goal), and there exist model identifiability issues [62]. In FLOW a trade-off has to be made between the computational cost and model expressiveness.

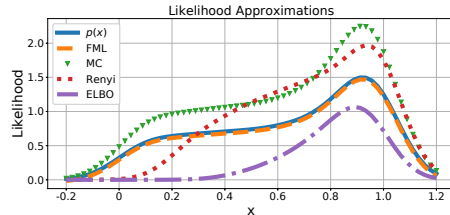

This paper presents a novel strategy for MLE learning for unnormalized statistical models, that allows efficient parameter estimation and accurate likelihood approximation. Importantly, while competing solutions can only yield stochastic upper/lower bounds, our treatment allows unbiased estimation of $\log$-likelihood and model parameters. Further, this setup can be used for effective sampling goals, and it has the ability to perform inference. This work makes the following contributions: ($i$) Derivation of a mini-max formulation of MLE, resulting in an unbiased $\log$-likelihood estimator directly amenable to stochastic gradient descent (SGD) optimization, with convergence guarantees. ($ii$) Amortized likelihood

Figure 1: Comparison of popular likelihood approximations: Monte-Carlo estimator (MC) (*e.g.*, contrastive divergence (CD) [33]), Renyi [40], importance-weighted ELBO [10], and the proposed FML. Cheap approximations often lead to biased estimate of likelihood, a point FML seeks to fix.

estimation with deep neural networks, enabling direct likelihood prediction and feature extraction (inference). ($iii$) Development of a novel training scheme for latent-variable models, presenting a competitive alternative to VI. ($iv$) We show that our models compare favorably to existing alternatives in likelihood-based distribution learning, both in terms of model estimation and sample generation.

## 2 Fenchel Mini-Max Learning

### 2.1 Preliminaries

**Maximum likelihood estimation** Given a family of parameterized probability density functions $\{p_\theta(x)\}_{\theta \in \Theta}$ and a set of empirical observations $\{x_i\}_{i=1}^n$, MLE seeks to identify the most probable model $\hat{\theta}_{\text{MLE}}$ via maximizing the expected model $log$-likelihood, *i.e.*, $\hat{\mathcal{L}}(\theta) \triangleq \frac{1}{n}\sum_{i=1}^n \log p_\theta(x_i)$. For flexible choices of $p_\theta(x)$, such as an unnormalized explicit-variable model $p_\theta(x) \propto \exp(-\psi_\theta(x))$ or latent variable model of the form $p_\theta(x) = \int p_\theta(x|z)p(z)dz$, direct optimization wrt MLE loss is typically computationally infeasible. Instead, relatively inexpensive likelihood approximations are often used to derive surrogate objectives.

**Variational inference**  Consider a latent variable model $p_\theta(x, z) = p_\theta(x|z)p(z)$. To avoid direct numerical estimation of $p_\theta(x)$, VI instead maximizes the variational lower bound to the marginal log-likelihood: $\text{ELBO}(p_\theta(x, z), q_\beta(z|x)) = \mathbb{E}_{q_\beta(z|x)} \log \left[ \frac{p_\theta(x,z)}{q_\beta(z|x)} \right]$, where $q_\beta(z|x)$ is an approximation to the true posterior $p_\theta(z|x)$. This bound tightens as $q_\beta(z|x)$ approaches the true posterior $p_\theta(z|x)$. For estimation, we seek parameters $\theta$ that maximize the ELBO, and the commensurately learned parameters $\beta$ are used in a subsequent inference task with new data. However, with such learning, samples drawn $x \sim p_{\hat{\theta}}(x|z)$ with $z \sim p(z)$ may not be as close to the training data as desired [12].

**Adversarial distribution matching**  Adversarial learning [25, 4] exploits the fact that many discrepancy measures have a dual formulation $\mathbb{D}(p_d, p_\theta) = \max_D \{V_{\mathbb{D}}(p_d, p_\theta; D)\}$, where $V_{\mathbb{D}}(p_d, p_\theta; D)$ is a variational objective that can be estimated with samples from the true distribution $p_d(x)$ and the model distribution $p_\theta(x)$, and $D(x)$ is an auxiliary function commonly known as the *critic* (or discriminator). To match draws from $p_\theta(x)$ to the data (sampled implicitly from $p_d(x)$) wrt $\mathbb{D}(p_d, p_\theta)$, one solves a mini-max game between the model $p_\theta(x)$ and critic $D(x)$: $p_\theta^* = \arg\min_{p_\theta} \{\max_D \{V_{\mathbb{D}}(p_d, p_\theta; D)\}\}$. In adversarial distribution matching, draws from $p_\theta(x)$ are often modeled via a deterministic function $G_\theta(z)$ that transforms samples from a (simple) source distribution $p(z)$ (*e.g.*, Gaussian) to the (complex) target distribution. This practice bypasses the difficulties involved when specifying a flexible yet easy to sample likelihood. However, it makes difficult the goal of subsequent likelihood estimation and inference of the latent $z$ for new data $x$.

<table>
<tr><td>

**Fenchel conjugacy**  Let $f(t)$ be a proper convex, lower-semicontinuous function; then its convex conjugate function $f^*(v)$ is defined as $f^*(v) = \sup_{t \in \mathcal{D}(f)} \{tv - f(t)\}$ where $\mathcal{D}(f)$ denotes the domain of function $f$ [34]. $f^*$ is also known as the *Fenchel conjugate* of $f$, which is again convex and lower-semicontinuous. The Fenchel conjugate pair $(f, f^*)$ are dual to each other, in the sense that $f^{**} = f$, *i.e.*, $f(t) = \sup_{v \in \mathcal{D}(f^*)} \{vt - f^*(v)\}$. As a concrete example, $(-\log(t), -1 - \log(-v))$ gives such a pair, as we exploit in the next section.

</td><td>

**Algorithm 1** Fenchel Mini-Max Learning

Empirical data distribution $\hat{p}_d = \{x_i\}_{i=1}^n$
Proposal $q(x)$, learning rate schedule $\{\eta_t\}$
Initialize parameters $\theta, b$
**for** $t = 1, 2, \cdots$ **do**
  Sample $\{x_{t,j}\}_{j=1}^m \sim \hat{p}_d(x)$, $\{x'_{t,j}\}_{j=1}^m \sim q(x)$
  $u_{t,j} = \psi(x_{t,j}) + b$,
  $I_{t,j} = \exp(\psi_\theta(x_{t,j}) - \psi_\theta(x'_{t,j}) - \log q(x'_{t,j}))$
  $J_t = \sum_j \{u_{t,j} + \exp(-u_{t,j})I_{t,j}\}$
  $[\theta, b] = [\theta, b] - \eta_t \nabla_{[\theta, b]} J_t$
  % Update proposal $q(x)$ if needed
**end for**

</td></tr>
</table>

**Biased finite sample Monte-Carlo for unnormalized statistical models**  For unnormalized statistical model $\tilde{p}_\theta(x) = \exp(-\psi_\theta(x))$, the naive Monte-Carlo estimator for the log-likelihood is given by $\log \hat{p}_\psi(x) = -\psi_\theta(x) - \log \hat{Z}_\theta$, where $\hat{Z}_\theta = \frac{1}{m} \sum_{j=1}^m \exp(-\psi_\theta(X'_j))$ is the finite-sample estimator for the normalizing constant $Z_\theta = \int e^{-\psi_\theta(x')} \, dx'$, with $\{X'_j\}$ i.i.d. uniform samples on $\Omega$. Via the Jensen's inequality (*i.e.*, $\mathbb{E}_X[\log f(X)] \leq \log(\mathbb{E}_X[f(X)])$), it is readily seen that $\mathbb{E}_{X'_{1:m}}[\log \hat{Z}_\theta] \leq \log(\mathbb{E}_{X'_{1:m}}[\hat{Z}_\theta]) = \log Z_\theta$, which implies the naive MC estimator gives an upper bound of the log-likelihood, *i.e.*, $\mathbb{E}_{X'_{1:m}}[\log \hat{p}_\psi(x)] \geq \log p_\psi(x)$. The inability to take infinite samples makes unbiased estimation of unnormalized statistical models a long-standing challenge posed to the statistical community, especially for high-dimensional problems [9].

## 2.2  Mini-Max formulation of MLE for unnormalized statistical models

For unnormalized statistical model $\tilde{p}_\theta(x) = \exp(-\psi_\theta(x))$, we rewrite model log-likelihood as

$$\log p_\theta(x) = \log \frac{e^{-\psi_\theta(x)}}{\int e^{-\psi_\theta(x')} \, dx'} = -\log \left( \int e^{\psi_\theta(x) - \psi_\theta(x')} \, dx' \right) \tag{1}$$

Recalling the Fenchel conjugate of $-\log(t)$, we have $-\log(t) = \max_u \{-u - \exp(-u)t + 1\}$, and the optimal value of $u$ is $u_t^* = \log(t)$. Plugging this into (1) yields the following expression

$$-\log p_\theta(x) = \min_{u_x} \left\{ u_x + \exp(-u_x) \int e^{\psi_\theta(x) - \psi_\theta(x')} \, dx' - 1 \right\}. \tag{2}$$

Since $u_x^* = \log \left( \int e^{\psi_\theta(x) - \psi_\theta(x')} \, dx' \right) = -\log p_\theta(x)$, we have $\exp(-u_x^*) = p_\theta(x)$. Consequently, the auxiliary dual variable $u$ is an estimate of the negative log-likelihood. The key insight here is that we have turned the numerical integration problem into an optimization problem. This may seem like a step backward at first sight, as we are still summing over the support and we have a dual variable to optimize. The payoff is that we can now sidestep the log term and estimate the log-likelihood in

an unbiased manner using finite MC samples, a major step-up over existing estimators. As argued below and verified experimentally, this extra optimization can be executed efficiently and robustly. This implies we are able to more accurately estimate unnormalized statistical models at a comparable budget, without compromising training stability.

Denote $I(x; \psi_\theta) = \int e^{\psi_\theta(x) - \psi_\theta(x')} \, \mathrm{d}x'$. To estimate $I(x; \psi_\theta)$ more efficiently, we may introduce a proposal distribution $q(x)$ with tractable likelihood and leverage an importance weighted estimator: $I(x; \psi_\theta) = \mathbb{E}_{X' \sim q}[\exp(\psi_\theta(x) - \psi_\theta(X') - \log q(X'))]$. We discuss the practical choice of proposal distribution in more detail in Section 2.4. Putting everything together, we have the following mini-max formulation of MLE for unnormalized statistical models:

$$\hat{\theta}_{\text{MLE}} = \arg\max_\theta \left\{ -\min_{\boldsymbol{u}} \left\{ \sum_i J_\theta(x_i; u_i, \psi) \right\} \right\}, \tag{3}$$

where $J_\theta(x; u, \psi) \triangleq u + \exp(-u)I(x; \psi_\theta)$.

In practice, we can model all $\{u_i\}$ with only one additional free parameter as $u_\theta(x) = \psi_\theta(x) + b_\theta$, where $b_\theta$ models the log-partition function, i.e., $b_\theta \triangleq \log Z_\theta$; we make explicit here that $u$ is a function of $\theta$, i.e., $u_\theta(x)$. Note that $b_\theta$ is the log-partition parameter to be learned, that minimizes the objective if and only if it equals the true log-partition. Although model parameters $\theta$ are shared between $u_\theta(x; b_\theta)$) and $\psi_\theta(x)$, they are fixed in the $u$-updates. Hence, when alternating between updating $\theta$ and $\boldsymbol{u}$ in (3), the update of $\boldsymbol{u}$ corresponds to refining the update of the log-partition function $b_\theta$ for fixed $\theta$, followed by updating $\theta$ with $b$ fixed; we have isolated learning the partition function (the $\min_u$ step) and the model parameters (the $\max_\theta$ step)[1]. We call this new formulation Fenchel Mini-Max Learning (FML), and summarize its pseudocode in Algorithm 1. For complex distributions, we also optimize the proposal $q(x)$ to enable efficient & robust learning with the importance weighted estimator.

Considering the form of $J(x; u, \psi_\theta)$, one may observe that the learning signal comes from contrasting data samples $x_i$ with a random draw $X'$ under the current model potential $\psi_\theta(x)$ (e.g., the term $\psi_\theta(x_i) - \psi_\theta(X')$). Figure 1 compares our FML to other popular likelihood approximation schemes. Unlike existing solutions, FML targets the exact likelihood without explicitly using finite-sample estimator for the partition function. Instead, FML optimizes an objective where the untransformed integral directly appears, which leads to an unbiased estimator provided the minimization is solved accurately.

### 2.3 Gradient analysis of FML

To further understand the workings of FML, we inspect the gradient of model parameters. In classical MLE learning, we have $\nabla \log p_\theta(x) = \frac{\nabla p_\theta(x)}{p_\theta(x)}$. That is to say, in MLE the gradient of the likelihood is normalized by the model evidence. A key observation is that, while $\nabla p_\theta(x)$ is difficult to compute, because of the partition function, we can easily acquire an unbiased gradient estimate of the inverse likelihood $\frac{1}{p_\theta(x)}$ using Monte-Carlo samples,

$$\nabla \left\{ \tfrac{1}{p_\theta(x)} \right\} = \nabla \left\{ \int \exp(\psi_\theta(x) - \psi_\theta(x')) \, \mathrm{d}x' \right\} = \int \nabla \{\exp(\psi_\theta(x) - \psi_\theta(x'))\} \, \mathrm{d}x' \tag{4}$$

which only differs from $\nabla \log p_\theta(x)$ by a factor of negative inverse likelihood

$$\nabla \left\{ \frac{1}{p_\theta(x)} \right\} = -\frac{\nabla p_\theta(x)}{(p_\theta(x))^2} = -\frac{\nabla \log p_\theta(x)}{p_\theta(x)}. \tag{5}$$

Now considering the gradient of FML, we have

$$\begin{aligned} \nabla J_\theta(x; \hat{u}_x, \psi) &= -\nabla \left\{ \exp(-\hat{u}_x) \int e^{\psi_\theta(x) - \psi_\theta(x')} \, \mathrm{d}x' \right\} \\ &= -\hat{p}_\theta(x) \nabla \left\{ \tfrac{1}{p_\theta(x)} \right\} = \frac{\hat{p}_\theta(x)}{p_\theta(x)} \nabla \log p_\theta(x) \approx \nabla \log p_\theta(x), \end{aligned} \tag{6}$$

where $\hat{u}_x$ denotes an approximate solution to the Fenchel maximization game (2) and $\hat{p}_\theta \triangleq \exp(-\hat{u}_x)$ is an approximation of the likelihood based on our previous analysis. We denote $\xi \triangleq \frac{\hat{p}_\theta(x)}{p_\theta(x)}$, and refer to $\log \xi$ as the approximation error. If this approximation $\hat{p}_\theta$ is sufficiently accurate then $\xi \approx 1$, which implies the FML gradient is a good approximation to the gradient of true likelihood.

When we model the auxiliary variable as $u(x) = \psi_\theta(x) + b$, then the FML gradient $\nabla J_\theta(x; u, \psi)$ differs from $\nabla \log p_\theta(x)$ by a common multiplicative factor $\xi = \exp(b - b_\theta)$ for all $x \in \Omega$. Next we show SGD is insensitive to this approximation error; FML still converges to the same solution of MLE even if $\xi$ deviates from 1 differently at each iteration.

## 2.4 Choice of proposal distribution

Like all importance-weighted estimators, the efficiency of FML critically depends on the choice of proposal $q(x)$. A poor match between the proposal and integrand can lead to extremely high variance [52], which compromises learning. In order to keep the variance in check, a general guiding principle for choosing a good $q(x)$ is to make it close to the data distribution $p_d$. Note this practice differs from the optimal minimal variance proposal, which is proportional to the integrand. However, it does not need to constantly update the proposal to adapt to the current parameter, which brings both robustness and computational savings. To obtain such a static proposal matched to the data distribution, we can pre-train a parameterized tractable sampler $q_\phi(x)$ with empirical data samples by maximizing the empirical model log-likelihood $\sum_i \log q_\phi(x_i)$, with $\phi$ parameterizing the proposal. Note that we only require the proposal $q(x)$ to be similar to the data distribution, using a rough approximation to facilitate the learning of an unnormalized model that more accurately characterize the data. The proposal does not necessarily need to capture every minute detail of the target distribution, as such simpler models are generally preferable for better computational efficiency, provided adequate approximation and coverage can be achieved. Popular choice of parameterized proposal include generative flows [53] or mixture of Gaussians [44]. We leave a more detailed specification of our treatment to the Supplementary Material (SUPP).

## 2.5 Convergence results

In modern machine learning, first order stochastic gradient descent (SGD) is a popular choice, and in many cases the only feasible approach, for large-scale problems. In the case of MLE, let $h(\theta; \omega)$ be an unbiased stochastic gradient estimator for $\hat{\mathcal{L}}(\theta)$, i.e., $\mathbb{E}_{\omega \sim p(\omega)}[h(\theta; \omega)] = \nabla \hat{\mathcal{L}}(\theta)$. Here we have used $\omega \sim p(\omega)$ to denote the source of randomness for $h(\theta; \omega)$. SGD finds a solution by using the following iterative procedure $\theta_{t+1} = \theta_t + \eta_t h(\theta_t; \omega_t)$, where $\{\eta_t\}$ is a pre-determined sequence commonly known as the learning-rate schedule and $\{\omega_t\}$ are iid draws from $p(\omega)$. Then under common technical assumptions on $h(\theta; \omega)$ and $\{\eta_t\}$, if there exists only one unique minimizer $\theta^*$ then the SGD solution $\hat{\theta}_{\text{SGD}} \triangleq \lim_{t \to \infty} \theta_t$ will converge to it [57].

Now consider FML's naive stochastic gradient estimator $\tilde{h}(\theta; \omega) = e^{-u(X)} \nabla \exp(\psi_\theta(X) - \psi_\theta(X'))$, where $X \sim \hat{p}_d, X' \sim \mathcal{U}(\Omega)$; the contrast $\psi_\theta(x) - \psi_\theta(x')$ between real and synthetic data is evident. Based on the analysis from the last section, we have the decomposition $\tilde{h}(\theta; \omega) = \xi \, h(\theta; \omega)$, where $h(\theta; \omega)$ is the unbiased stochastic gradient term and $\xi$ relates to the (unknown) approximation error. Using the same learning rate schedule, we are updating model parameter with effective random step-sizes $\tilde{\eta}_t \triangleq \xi_t \eta_t$ relative to SGD with MLE, where $\xi_t$ depends on the current approximation error. We formalize this as the generalized SGD problem described below.

**Problem 2.1** (Generalized SGD). Let $h(\theta; \omega), \omega \sim p(\omega)$ be an unbiased stochastic gradient estimator for objective $f(\theta)$, $\{\eta_t > 0\}$ is the fixed learning rate schedule, $\{\xi_t > 0\}$ is the random perturbations to the learning rate. We want to solve for $\nabla f(\theta) = 0$ with the iterative scheme $\theta_{t+1} = \theta_t + \tilde{\eta}_t \, h(\theta_t; \omega_t)$, where $\{\omega_t\}$ are iid draws and $\tilde{\eta}_t = \eta_t \xi_t$ is the randomized learning rate.

**Proposition 2.2** (Generalized stochastic approximation). Under the standard regularity conditions listed in Assumption D.1 in the SUPP, we further assume $\sum_t \mathbb{E}[\tilde{\eta}_t] = \infty$ and $\sum_t \mathbb{E}[\tilde{\eta}_t^2] < \infty$. Then $\theta_n \to \theta^*$ with probability 1 from any initial point $\theta_0$.

*Remark.* This is a straightforward generalization of the Robbins-Monro theory. The original proof still applies by simply replacing expectation wrt the deterministic sequence $\{\eta_t\}$ with the randomized sequence $\{\tilde{\eta}_t\}$. Assumptions $\sum_t \mathbb{E}[\tilde{\eta}_t] = \infty$ and $\sum_t \mathbb{E}[\tilde{\eta}_t^2] < \infty$ can be satisfied by saying $\{\log \xi_t\}$ is bounded. The $u$-updates used in FML force $\{\log \xi_t\}$ to stay close to zero, thereby enforcing the boundedness condition. Although such assumptions are too strong for deep neural nets, empirically FML converges to very reasonable solutions. We discuss more general theories in the SUPP.

**Corollary 2.3.** Under the assumptions of Prop. 2.2, FML converges to $\hat{\theta}_{\text{MLE}}$ with SGD.

# 3 FML for Latent Variable Models and Sampling Distributions

## 3.1 Likelihood-free modeling & latent variable models

One can reformulate generative adversarial networks (GANs) [25, 30] into a latent-variable model, by introducing arbitrarily small Gaussian perturbations. Specifically, $X' = G_\theta(Z) + \sigma\zeta$, where

$\zeta \sim \mathcal{N}(0, 1)$ is standard Gaussian, and $\sigma$ is the noise standard deviation. This gives the joint likelihood $p_\theta^\dagger(x, z) = \mathcal{N}(G_\theta(z), \sigma^2)p(z)$. It is well known the marginal likelihood $p_\theta^\dagger(x)$ converges to $p_\theta(x)$ as $\sigma$ goes to zero [4]. As such, we can always use a latent-variable model to approximate the likelihood of an implicitly defined distribution $p_\theta(x)$, which is easy to sample from. It also allows us to associate generator parameters $\theta$ to likelihood-based losses.

## 3.2 Fenchel reformulation of marginal likelihood

Replacing the $\log$ term with its Fenchel dual, we have the following alternative expression for the marginal likelihood: $\log p_\theta(x) = \log(\int p_\theta(x, z) \, \mathrm{d}z) = \min_{u_x}\{u_x + \exp(-u_x)I(x; p_\theta) - 1\}$, where $I(x; p_\theta) \triangleq \int p_\theta(x, z) \, \mathrm{d}z$. Note that, different from the last section, here estimate $\hat{u}_x$ provides a direct approximation to the marginal likelihood $\log p_\theta(x)$ rather than its negative. By analogy with variational inference (VI), an approximate posterior $q_\beta(z|x)$ can also be introduced, assuming the role of proposal distribution for the integral term. Model parameter $\theta$ can be learned via the following mini-max setup

$$\max_\theta\{\min_{\boldsymbol{u}}\{\underbrace{\mathbb{E}_{X \sim p_d}[u_X + \exp(-u_X)I(X; p_\theta, q_\beta)]}_{\mathcal{J}(\boldsymbol{u}; p_\theta, q_\beta)}\}\}, \tag{7}$$

where $I(x; p_\theta, q_\beta) \triangleq \mathbb{E}_{q_\beta}[\frac{p_\theta(x, Z)}{q_\beta(Z|x)}]$ is the importance weighted estimator with proposal $q_\beta(z|x)$, and $\boldsymbol{u} \in \mathbb{R}^n$ is a vector modeling the marginal likelihood $\log p_\theta(x_i)$ for each training example $x_i$ with $u_i$. A good proposal encodes the association between $x$ and $z$ (this is expanded upon in the SUPP); as such, we also refer to $q_\beta$ as the inference distribution. We will return to the optimization of inference parameter $\beta$ in Section 3.3. Our analysis from Sections 2.3 to 2.5 also applies in the latent variable case and is not repeated here. To further stabilize the training, annealed training can be considered, replacing integrand $\frac{p_\theta(x, z)}{q_\beta(z|x)}$ with $\frac{p_\theta^{\tau_t}(x|z)p(z)}{q_\beta(z|x)}$ as in Neal [49]. Here $\{\tau_t\}$ is the annealing schedule, monotonically increasing wrt time $t$ going from $\tau_0 = 0$ to $\tau_\infty = 1$.

## 3.3 Optimization of inference distribution

The choice of proposal distribution $q_\beta(z|x)$ is important for the statistical efficiency of FML. To address this issue, we propose to encourage more informative proposal via regularizing the vanilla FML objective. In particular, we consider regularizing with the mutual information $I_p \triangleq \mathbb{E}_p[\log \frac{p(X, Z)}{p(X)p(Z)}]$.

Let us denote our model distribution $p_\theta(x, z)$ as $\rho$ and the approximate joint $q_\beta(x, z) \triangleq q_\beta(z|x)p_d(x)$ as $q$, and the respective mutual information are denoted as $I_\rho$ and $I_q$. It is necessary to regularize both $I_\rho$ and $I_q$, since $I_q$ directly encourage more informative proposal, while the "informativeness" is upper bounded by $I_\rho$ [2]. In other words, this encourages the proposal to approach the posterior.

Direct estimation of $I_\rho$ and $I_q$ is infeasible, due to the absence of analytical expressions for the marginals $p_\theta(x)$ and $q_\beta(z)$. Instead we use their respective lower bounds [5, 2] $\mathcal{D}_\rho(\theta, \beta) \triangleq \mathbb{E}_{(X,Z) \sim p_\theta}[\log q_\beta(Z|X)]$ and $\mathcal{D}_q(\beta|\theta) \triangleq \mathbb{E}_{(X,Z) \sim q_\beta}[\log p_\theta(X|Z)]$ as our regularizer (see the SUPP for details). Note these bounds are tight as the proposal $q_\beta(z|x)$ approaches the true posterior $p_\theta(z|x)$ (Lemma 5.1, Chen et al. [13]). We then solve the following regularized mini-max game

$$\max_{\theta, \beta}\left\{\min_{\boldsymbol{u}}\{\mathcal{J}(\boldsymbol{u}, \theta, \beta)\} - \lambda_q \mathcal{D}_q(\beta|\theta) - \lambda_\rho \mathcal{D}_\rho(\theta, \beta)\right\}. \tag{8}$$

Here the nonnegative $\{\lambda_\rho, \lambda_q\}$ are the regularization strengths, and we have used notation $\mathcal{D}_q(\beta|\theta)$ to highlight the fact this term does not contribute to the gradient of model parameter $\theta$. Solving (8) using standard simultaneous gradient descent/ascent as in standard GAN training is observed to be efficient and stable in practice.

## 3.4 Amortized inference of marginal likelihoods

Unlike the explicit likelihood case from Section 2, the marginal likelihoods $\{\log p_\theta(x_i)\}$ are no longer directly related by an explicit potential function $\psi_\theta(x)$. Individually update $u_i$ for each sample $x_i$ is computationally inefficient: ($i$) it does not scale to large datasets; ($ii$) parameters are not shared across samples; ($iii$) it does not permit efficient prediction of the likelihood at test time for a new observation $x_{\text{new}}$. Motivated by its success in variational inference, we propose to employ the amortization technique to tackle the above issues [14]. When optimizing some objective function with distinct parameters $\zeta_i$ associated with each training example $x_i$, e.g., $\mathcal{L}(\theta, \boldsymbol{\zeta}) = \sum_i \ell_\theta(x_i; \zeta_i)$, amortized learning replaces these parameters with a parameterized function $\zeta_\phi(x)$ with $\phi$ as the amortization parameters. The optimization is then carried out wrt the amortized objective $\mathcal{L}(\theta, \phi) = \sum_i \ell_\theta(x_i; \zeta_\phi(x_i))$ instead. Contextualized under our FML, we amortize the marginal likelihood estimate $\{u_i\}$ with

a parameterized function $u_\phi(x)$, and optimize $\max_\theta\{\min_\phi\{\mathbb{E}_{X\sim p_d}[\mathcal{J}(u_\phi; p_\theta, q_\beta)]\}\}$ instead of (7). Since $\mathbb{E}_{p_d}[\log p_\theta] = \min_u\{\mathbb{E}_{p_d}[\mathcal{J}(u_X; p_\theta, q_\beta)]\} \leq \min_\phi\{\mathbb{E}_{p_d}\mathcal{J}(u_\phi(x); p_\theta, q_\beta)\}$, amortized latent FML effectively optimizes an upper bound of the likelihood loss. This bound tightens as the function family $u_\phi$ becomes more expressive, which makes expressive deep neural networks an appealing choice for $u_\phi$ [35]. To further improve parameter efficiency, we note parameter $\phi$ can be shared with the proposal parameter $\beta$ used by $q_\beta(z|x)$.

### 3.5 Sampling From Unnormalized Distribution

There are problems for which we are given an unnormalized distribution $p_{\psi^*}(x) \propto \exp[-\psi^*(x)]$ and no data samples; we would like to model $p_{\psi^*}(x)$ in the sense that we'd like to efficiently sample from it. This problem arises, for example, in reinforcement learning [31], among others. To address this problem under FML, we propose to parameterize a sampler $X = G_\theta(Z), Z \sim p(z)$ and a nonparametric potential function $\psi_\theta(x)$ [2]. FML is used to estimate the model likelihood via solving

$$\max_\psi\{-\min_b\{\mathcal{F}(\psi, b; \theta)\}\}, \ \mathcal{F}(\psi, b; \theta) \triangleq \mathbb{E}_{Z\sim p(z)}[\mathcal{J}(G_\theta(Z), u_{\psi,b}, \psi)] \tag{9}$$

where $u_{\psi,b}(x) = \psi_\theta(x) + b$ is our estimate for $-\log p_\theta(x)$ implicitly defined by $G_\theta(z)$.

To match model samples to the target distribution, $G_\theta(z)$ is trained to minimize the KL-divergence

$$\text{KL}(p_\theta \| p_{\psi^*}) = \mathbb{E}_{X\sim p_\theta}[\log p_\theta(X) - \log p_{\psi^*}(X)] = \mathbb{E}_{X\sim p_\theta}[\log p_\theta(X) + \psi^*(X)] + \log Z_{\psi^*}$$

Since the last term is independent of model parameter $\theta$, we obtain the KL-based training objective $\mathcal{J}_{\text{KL}}(\theta; \psi, b, \psi^*) \triangleq \mathbb{E}_{Z\sim p(z)}[\psi^*(G_\theta(Z)) - u_{\psi,b}(G_\theta(Z))]$ by replacing $\log p_\theta(x)$ with our FML estimate. Due to the dependence of $u_b(x)$ on $\theta$, the final learning procedure is

$$[\psi_t, b_t] = [\psi_{t-1}, b_{t-1}] - \eta_t \nabla_{[\psi, b]}\mathcal{F}(\psi_{t-1}, b_{t-1}; \theta_t), \ \theta_{t+1} \leftarrow \theta_t - \eta_t \nabla_\theta \mathcal{J}_{\text{KL}}(\theta_t; \psi_t, b_t, \psi^*).$$

## 4  Related Work

**Fenchel duality** In addition to optimization schemes, the Fenchel duality also finds successful applications in probabilistic modeling. Prominent examples include divergence minimization [3] and likelihood-ratio estimation [50], and more recently adversarial learning [51]. In discrete learning, Fagan and Iyengar [20] employed it to speedup extreme classification. To the best of the authors' knowledge, Fenchel duality has not been applied previously to likelihoods with latent variables.

**Nonparametric density estimation** To combat the biased estimation of the partition function, Burda et al. [9] proposed a conservative estimator, which partly alleviates this issue. Parallel to our work, Dai et al. [16] explored Fenchel duality in the setting of MLE for an unnormalized statistical model estimation, under the name dynamics dual embedding (DDE), which seeks optimal embedding in the space of probability measures. The authors used parameterized Hamiltonian flows for distribution embeddings, which limits its scalability and expressiveness. In particular, DDE fails if the search space does not contain the target distribution, while our formulation only requires the support of the proposal distribution to cover that of the target.

**Adversarial distribution learning** The proposed FML framework is complementary to the development of GANs. FML prioritizes the learning of a potential function, while GANs have focused on the training of a sampler. Both schemes are derived via *learning by contrast*. Notably $f$-GANs contrast the difference between likelihoods under respective models, while our FML contrasts data samples with proposal samples under the current model potential. Synergies can be explored between the two schemes.

**Approximate inference** Compared with VI, FML optimizes a direct estimate of the marginal likelihood instead of a variational bound. While tighter bounds can be achieved for VI via importance re-weighting [10], flexible posteriors [47] and alternative evidence scores [62], these strategies do not necessarily improve performance [55]. Another fundamental difference is that while VI discards all conditional likelihoods after the ELBO evaluation, FML consolidates them into an estimate of the marginal likelihood through SGD.

**Sampling unnormalized potentials** This is one of the fundamental topics in statistics and computer science [45]. Recent studies have explored the use of deep neural sampler for this purpose: Feng et al. [21] trains the sampler with kernel Stein variational gradients, and Li et al. [38] adversarially updates the sampler based on the adaptive contrast technique [47]. FML provides an expressive, scalable and

Table 1: Quantitative evaluation on toy models.

| Model | Parameter estimation error [†] ↓ | | | | | Likelihood consistency score ↑ | | | | |
|---|---|---|---|---|---|---|---|---|---|---|
| | banana | kidney | rings | river | wave | banana | kidney | rings | river | wave |
| MC | 3.46 | 3.9 | 4.71 | 1.71 | 1.78 | 0.961 | 0.881 | 0.508 | 0.702 | 0.619 |
| SM [36] | 7.79 | 2.75 | 3.62 | 1.64 | 2.61 | × | × | × | × | × |
| NCE [28] | 3.88 | 2.5 | 4.81 | 2.85 | **1.20** | 0.968 | 0.882 | 0.557 | 0.721 | 0.759 |
| KEF [59] | × | × | × | × | × | 0.973 | 0.755 | 0.183 | 0.436 | 0.265 |
| DDE [16] | 6.59 | 7.31 | 24.9 | 29.1 | 25.7 | 0.944 | 0.830 | 0.426 | 0.520 | 0.186 |
| FML (ours) | **3.05** | **1.9** | **2.59** | **1.13** | 1.27 | **0.974** | **0.901** | **0.562** | **0.731** | **0.782** |

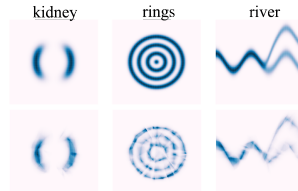

Figure 2: FML predicted likelihood using nonparametric potentials.

numerically stable solution based on the simulation of a Langevin gradient flow.

# 5 Experiments

To validate the proposed FML framework and benchmark it against state-of-the-art methods, we consider a wide range of experiments, using synthetic and real-world datasets. All experiments are implemented with Tensorflow and executed on a single NVIDIA TITAN X GPU. Details of the experimental setup are provided in the SUPP, due to space limits, and our code is from `https://www.github.com/chenyang-tao/FML`. For the evaluation metrics reported, ↑ indicates a higher score is considered better, and *vice versa* with ↓. Our goal is to verify FML works favorably or similarly compared with competing solutions under the same setup, not to beat state-of-the-art results.

## 5.1 Estimating unnormalized statistical models

We compare FML with competing solutions on parameter estimation and likelihood prediction with unnormalized statistical models. We report × if a method is unable to compute or failed to reach a reasonable result. Grid search is used for KDE to optimize the kernel bandwidth.

Table 2: log-likelihood evaluation on UCI datasets ↑.

| Model | wine-red | wine-white | yeast | htru2 |
|---|---|---|---|---|
| KDE | 7.74 | 7.74 | 3.01 | 15.47 |
| GMM | 7.42 | 7.97 | 4.82 | 22.06 |
| DDE | 7.45 | 7.18 | 3.79 | 18.83 |
| FLOW | 7.09 | 7.75 | 3.31 | 20.48 |
| NCE | 7.29 | 7.98 | 4.84 | 22.05 |
| FML | **8.45** | **8.20** | **4.96** | **22.15** |

**Parameter estimation for unnormalized models** We first benchmark the performance on parameter estimation with a number of representative toy models, including both continuous distributions with varying dimensionality (see SUPP for details). The exact parametric form of the potential function is given, and the task is to estimate the parameter values that generate the samples. We use 1,000 and 5,000 samples, respectively, for training and evaluation. To assess performance, we repeat each experiment 10 times and report the mean absolute error $\|\hat{\theta} - \theta^*\|_1$, where $\hat{\theta}$ and $\theta^*$ denote the parameter estimate and ground-truth, respectively. We benchmark FML against naive Monte-Carlo, score matching, noise contrastive estimation and dual dynamics embedding, with results reported in Table 1. FML provides comparable, if not better, performance on all the models considered.

**Nonparametric likelihood prediction** In the absence of an explicit parametric model of the likelihood, a deep neural network is used as a nonparametric model of the potential. To evaluate model performance, we consider the likelihood consistency score, defined as the correlation between the learned nonparametric potential and the ground truth potential, *i.e.*, $\mathrm{corr}(\log p_{\theta^*}(X), \log p_{\hat{\theta}}(X))$, where the expectation is taken wrt ground-truth samples. The results are summarized in Table 1. In Figure 2, we also visualize the nonparametric FML estimates of the likelihood compared with ground truth. Note SM proved computationally unstable in all cases, and DDE has to be trained with a smaller learning rate, due to stability issues.

In addition to the toy examples, we also evaluate the proposed FML on real datasets from the UCI data repository [17]. To evaluate model performance, we randomly split the data into ten folds, and use seven of them for training and three of them for evaluation. To cope with the high-dimensionality of the data, we use a GMM proposal for both NCE and FML. The averaged $log$-likelihood on the test set is reported in Table 2, and the proposed FML shows an advantage over its counterparts.

## 5.2 Latent variable models and generative modeling

Our next experiment considers FML-based training for latent variable models and generative modeling tasks. In particular, we directly benchmark FML against the VAE [37], for modeling complex distributions, such as images and natural language, for real-world applications. We focus on evaluating the model's ability to (efficiently) synthesize realistic samples. Additionally, we also demonstrate how FML can assist the training of generative adversarial nets by following the variational annealing setup

Table 3: VAE quantitative results.

| MNIST | IS↑ | FID↓ | $-\log \hat{p} \downarrow$ |
|---|---|---|---|
| VAE | 8.08 | 24.3 | 103.7 |
| FML | **8.30** | **22.7** | **101.5** |

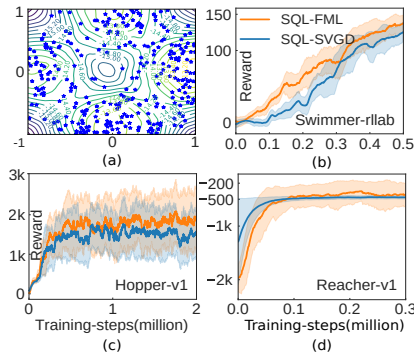

Figure 3: Sampled images from FML-trained models.

described in Tao et al. [63], with results summarized in Table 4. Our FML-based solution outperforms DAE score estimator [1] based DFM [64] in terms of FID, while giving similar performance in IS.

**Image datasets**  We applied FML-training to a number of popular image datasets including MNIST, CelebA, and Cifar10. The following metrics are considered for quantitative evaluation of model performance: ($i$) *Inception Score* (IS) [58], ($ii$) *Fréchet Inception Distance* (FID) [32], and ($iii$) negative $log$-likelihood estimates [65]. See Table 3 for quantitative evaluations (additional results on CelebA see SUPP), and images sampled from the FML-trained models are presented in Figure 3 for qualitative assessment. FML-based training consistently improves model performance wrt quantitative measures, which is also verified based on our human evaluation (see SUPP).

Table 4: GAN quantitative results.

| Cifar10 | IS↑ | FID↓ |
|---|---|---|
| GAN | 6.29 | 37.4 |
| DFM | **6.93** | 30.7 |
| FML | 6.91 | **30.0** |

**Natural language models**  We further apply FML to the learning of natural language models. The following two benchmark datasets are considered: ($i$) *EMNLP WMT news* [26] and ($ii$) *MS COCO* [43]. In accordance with standard literature in language modeling, we report both *perplexity* (PPL) [8] and *BLEU* [54] scores. Note PPL is an evaluation metric based on the likelihood. Quantitative results along with sentence samples generated from trained models are reported in Table 5. FML-based training leads to consistently improved performance wrt both PPL and BLEU; it also typically generates more coherent sentences compared with its counterpart.

Table 5: Results on language models, with the example synthesized text representative of typical results.

|  | PPL ↓ | BLEU-2 ↑ | BLEU-3 ↑ | BLEU-4 ↑ | BLEU-5 ↑ |
|---|---|---|---|---|---|
| *EMNLP WMT news* | | | | | |
| VAE | 12.5 | 76.1 | 46.8 | 23.1 | 11.6 |
| FML | **11.6** | **77.2** | **47.4** | **24.3** | **12.2** |
| *MS COCO* | | | | | |
| VAE | 9.5 | 82.1 | 60.7 | 38.9 | 24.8 |
| FML | **8.6** | **84.2** | **64.4** | **40.3** | **25.2** |

Sampled sentences from respective models on *WMT news*

VAE *"China's economic crisis, the number of US exports, which is still in recent years of the UK' s population."*

FML *"In addition, police officials have also found a new investigation into the area where they could take a further notice of a similar investigation into."*

### 5.3 Sampling unnormalized distributions

Our final experiment considers an application in reinforcement learning (RL) with FML-trained neural sampler. We benchmark the effectiveness of our FML-based sampling scheme described in Sec 3.5 by comparing it with the SVGD sampler used in state-of-the-art soft Q-learning implementation [31]. We examine the performance on three continuous control tasks, namely *swimmer, hopper and reacher*, defined in *OpenAI gym* [7] and *rllab* [18] environments, with results summarized in Figure 4. Figure 4(a) overlays samples from the FML-trained policy network on the potential of the model estimated optimal policy, verifying FML's capability to capture complex multi-modal distributions. The evolution of policy rewards wrt training iterations is provided in Figure 4(b-d), and FML-based policy updates improve on original SVGD updates.

Figure 4: Soft Q-Learning with FML.

### 6 Conclusion

We have developed a scalable and flexible learning scheme for probabilistic modeling. Rooted in classical MLE learning, our solution handles inference, estimation, sampling and likelihood evaluation in a unified framework, without major compromises. Empirical evidence verified the proposed method delivers competitive performance on a wide range of tasks.

## Acknowledgements

The authors would like to thank the anonymous reviewers for their insightful comments. This research was supported in part by DARPA, DOE, NIH, ONR, NSF and RTI Internal research & development funds. J Chen was partially supported by China Scholarship Council (CSC). W Lu and J Feng were supported by the Shanghai Municipal Science and Technology Major Project and ZJ Lab (No. 2018SHZDZX01).

## Footnotes

[1]In practice, we find that instead of separated updates, simultaneous gradient descent of $\theta$ and $b$ also works well.

[2]With slight abuse of notation, we assume $\psi_\theta(x)$ is parameterized by $\psi$ to avoid notation collision with sampler $G_\theta(z)$.

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
