[Supplementary Material · fml-supp.pdf]

# Supplementary Material for "On Frenchel Mini-Max Learning"

**Chenyang Tao** [1]  **Liqun Chen** [1]  **Shuyang Dai** [1]  **Junya Chen** [1 2]  **Bai Ke** [1]  **Dong Wang** [1]  **Jianfeng Feng** [3]
**Wenlian Lu** [2]  **Georgiy Bobashev** [4]  **Lawrence Carin** [1]

## A. Notations and Assumptions

Capital letters ($e.g.$, $X$) are used to denote random variables, and $\mathbb{E}_{X \sim p}[f(X)]$ denotes the expectation of function $f(x)$ wrt distribution $p(x)$. $\nabla f_\theta$ denotes the gradient of function $f_\theta(x)$ wrt parameters $\theta$, gradient wrt spatial parameters will be denoted as $\nabla_x f_\theta$. $\Delta_x$ is used to denote the Laplacian operator wrt spatial variable $x$. We say $p_\theta(x)$ is an *explicit* likelihood if its evaluation does not involve marginalization over latent variables. To simplify discussions, we always assume continuous variables, and probability measures of interest are defined on a compact domain $\Omega \subset \mathbb{R}^d$. Without loss of generality, $\Omega$ is assumed to have unit volume.

## B. Comparison of Popular Probabilistic Modeling Procedures

We summarized popular probabilistic modeling schemes' strength wrt the goals of inference, sampling, likelihood evaluation and scalability in Table S1. Note that this table is composed based on standard understand of these methods. For some specialized applications some of these methods can be extended beyond the limitations outlined in this table.

## C. Biased Likelihood Estimation with Finite Sample Monte-Carlo Estimation

Consider the following naive Monte-Carlo estimator for the log-likelihood

$$\log \hat{p}_\psi(x) = -\psi_\theta(x) - \log \hat{Z}_\theta, \tag{1}$$

where

$$\log \hat{Z}_\theta = \log \left( \frac{1}{m} \sum_{j=1}^{m} \exp(-\psi_\theta(X'_j)) \right), \tag{2}$$

---

[*]Equal contribution   [1]Electrical & Computer Engineering, Duke University, Durham, NC, USA [2]School of Mathematical Science, Fudan University, Shanghai, China [3]ISTBI, Fudan University, Shanghai, China [4]RTI International, Research Triangle Park, NC, USA. Correspondence to: Chenyang Tao <chenyang.tao@duke.edu>, Lawrence Carin <lcarin@duke.edu>.

*Table S1.* Comparison of popular probabilistic modeling procedures.

| Model | Inference | Sampling | Likelihood | Scalability |
|---|---|---|---|---|
| FML (ours) | Yes | Yes | Estimate | Good |
| CD | Yes | Yes | No | Good |
| SM | No | No | No | Poor |
| NCE | No | No | Estimate | Depends |
| KEF | No | No | No | Poor |
| DDE | No | Yes | Exact | Low |
| VI | Yes | Yes | Bound | Good |
| Flow | No | Yes | Exact | Tricky |
| Stein | No | Yes | No | Medium |
| GAN | No | Yes | No | Good |

is the finite sample estimator for the log-partition, with $X'_j$ sampled uniformly from $\Omega$. Via the Jensen inequality ($i.e.$, $\mathbb{E}_X[\log f(X)] \leq \log(\mathbb{E}_X[f(X)])$), it is easy to see

$$\mathbb{E}_{X_j}[\log \hat{Z}_\theta] \leq \log(\mathbb{E}_{X_j}[\hat{Z}_\theta]) = \log Z_\theta, \tag{3}$$

which implies the naive MC estimator actually gives an upper bound of the log-likelihood

$$\mathbb{E}_{X_j}[\log \hat{p}_\psi(x)] \geq \log p_\psi(x). \tag{4}$$

To partly alleviate this issue, **?** considered an alternative estimator that lower bounds the log-likelihood.

## D. Technical Assumptions for Robbins-Monro Stochastic Approximation

**Assumption D.1.** (Standard regularity conditions for Robbins-Monro stochastic approximation).

$A1.$ $h(\theta) \triangleq \mathbb{E}_\omega[h(\theta; \omega)]$ is Lipschitz continuous;

$A2.$ The ODE $\dot{\theta} = h(\theta)$ has a unique equilibrium point $\theta^*$, which is globally asymptotically stable;

$A3.$ The sequence $\{\theta_t\}$ is bounded with prob 1;

$A4.$ The noise sequence $\{\omega_t\}$ is a martingale difference sequence;

$A5.$ For some finite constants $A$ and $B$ and some norm $\| \cdot \|$ on $\mathbb{R}^d$, $\mathbb{E}[\|\omega_t\|^2] \leq A + B\|\theta_t\|^2$ a.s. $\forall t \geq 1$.

*Remark.* In the context of stochastic optimization, the globally asymptotic stability can be implied, for example, when $f(\theta)$ is strict convex (recall $h(\theta) = \nabla f(\theta)$).

## E. Proof of Proposition 3.2

*Proof.* We only need to verify that convergence still holds in probability when Robbins-Monro condition is satisfied in expectation. Without loss of generality suppose

$$\langle \theta - \theta^*, h(\theta) \rangle \le 0 \tag{5}$$

holds for all $\theta$. Define a Markov chain $\theta_t$ by taking $\theta_1$ to be an arbitrary constant and define

$$\theta_{t+1} - \theta_t = -\xi_t \eta_t h(\theta_t). \tag{6}$$

Let $b_n = \mathbb{E}\left[\|\theta_t - \theta^*\|_2^2\right]$. We shall find conditions under which $\lim_{n \to \infty} b_n = 0$, no matter what the initial value $\theta_1$, which implies the convergence in probability of $\theta_t$ to $\theta^*$. From (6), we have:

$$
\begin{aligned}
b_{n+1} &= \mathbb{E}\left[\|\theta_{t+1} - \theta^*\|_2^2\right] = \mathbb{E}\left[\|(\theta_{t+1} - \theta_t) + (\theta_t - \theta^*)\|_2^2\right] \\
&= b_n + 2\mathbb{E}\left[\langle \theta_{t+1} - \theta_t, \theta_t - \theta^* \rangle\right] + \mathbb{E}\left[\|\theta_{t+1} - \theta_t\|_2^2\right] \\
&= b_n - 2\mathbb{E}\left[\xi_t \eta_t\right]\mathbb{E}\left[\langle \theta_t - \theta^*, h(\theta_t) \rangle\right] \\
&\quad + \mathbb{E}[\xi_t^2 \eta_t^2]\mathbb{E}\left[\|h(\theta_t)\|_2^2\right]
\end{aligned}
$$

Set $d_n = \mathbb{E}\left[\langle \theta_t - \theta^*, h(\theta_t) \rangle\right]$, $e_t = \mathbb{E}\left[\|h(\theta_t)\|_2^2\right]$, we can write

$$b_{n+1} - b_n = -2\mathbb{E}[\xi_t^2 \eta_t^2]d_n + \mathbb{E}[\xi_t \eta_t]e_t. \tag{7}$$

Note from (5), $d_n \ge 0$, while from the assumption on function $h(\cdot)$, $0 \le e_t < \infty$, together with $\sum \mathbb{E}\left[\xi_t^2 \eta_t^2\right] < \infty$, we have $\sum \mathbb{E}\left[\xi_t^2 \eta_t^2\right] e_t$ converges. Summing (7) gives

$$b_{t+1} = b_1 + \sum_{j=1}^{t} \mathbb{E}\left[\xi_j^2 \eta_j^2\right] e_j - 2\sum_{j=1}^{t} \mathbb{E}\left[\xi_j \eta_j\right] d_j \tag{8}$$

Since $b_{n+1} \ge 0$, we obtain

$$\sum_{j=1}^{t} \mathbb{E}\left[\xi_j \eta_j\right] d_j \le \frac{1}{2}\left[b_1 + \sum_{j=1}^{t} \mathbb{E}\left[\xi_j^2 \eta_j^2\right] e_j\right] < \infty$$

It follows from (8) that $\lim_{n \to \infty} b_n = b$ exists. $b$ equals to 0 is proved in Robbins and Monro's paper, more details can be found in (Robbins & Monro, 1951) . $\square$

## F. Proof of Corollary 3.3

*Proof.* Based on analysis from Section 3.2, FML executes SGD wrt MLE gradient with randomly perturbed step size $\tilde{\eta}_t$. Then result directly follows from Proposition 3.1. $\square$

## G. More general results on FML convergence

The following results relaxes the strong assumptions on the uniqueness of global minimizer, proving that under SGD FML training also reaches a stationary point of the ground-truth likelihood function as standard MLE training does. This results applies more generally to modern learning frameworks such as deep neural net. We note that due to the stochasticity and the nonlinearity involved, both FML and MLE may reach different solutions in separate runs for deep nets.

**Assumption G.1.** (Weaker regularity conditions for generalized Robbins-Monro stochastic approximation).

$B1$. The objective function $f(\theta)$ is second-order differentiable.

$B2$. The objective function $f(\theta)$ has a Lipschitz-continuous gradient, i.e., there exists a constant $L$ satisfying

$$-LI \preceq \nabla^2 f(\theta) \preceq LI,$$

$B3$. The noise has a bounded variance, i.e., there exists a constant $\sigma > 0$ satisfying $\mathbb{E}\left[\|h(\theta_t; \omega_t) - \nabla f(\theta_t)\|^2\right] \le \sigma^2$.

**Theorem G.2.** *Under the technical conditions listed in Assumption G.1, the SGD solution $\{\theta_t\}_{t>0}$ updated with generalized Robbins-Monro sequence ($\tilde{\eta}_t$: $\sum_t \mathbb{E}[\tilde{\eta}_t] = \infty$ and $\sum_t \mathbb{E}[\tilde{\eta}_t^2] < \infty$) converges to a stationary point of $f(\theta)$ with probability 1 (equivalently, $\mathbb{E}\left[\|\nabla f(\theta_t)\|^2\right] \to 0$ as $t \to \infty$).*

*Proof.* Define a Markov chain $\theta_t$ by taking $\theta_1$ to be an arbitrary constant vector:

$$\theta_{t+1} - \theta_t = -\xi_t \eta_t h(\theta_t; \omega_t) \triangleq -\tilde{\eta}_t h_t(\theta_t)$$

where $\tilde{\eta}_t = \xi_t \eta_t$ and $h_t(\theta_t) = h(\theta_t; \omega_t)$.

By Taylor's theorem, the objective will be

$$
\begin{aligned}
f(\theta_{t+1}) &= f(\theta_t - \tilde{\eta}_t h_t(\theta_t)) \\
&= f(\theta_t) - \tilde{\eta}_t h_t(\theta_t)^\top \nabla f(\theta_t) + \frac{\tilde{\eta}_t^2}{2} h_t(\theta_t)^\top \nabla^2 f(\theta_t) h_t(\theta_t)
\end{aligned}
$$

Taking the expected value,

$$
\begin{aligned}
\mathbb{E}\left[f(\theta_{t+1})\big|\theta_t\right] &\le f(\theta_t) - \mathbb{E}\left[\tilde{\eta}_t\right]\mathbb{E}\left[h_t(\theta_t)^\top \nabla f(\theta_t)\big|\theta_t\right] \\
&\quad + \frac{L}{2}\mathbb{E}[\tilde{\eta}_t^2]\mathbb{E}\left[\|h_t(\theta_t)\|^2\big|\theta_t\right] \\
&\le f(\theta_t) - \left(\mathbb{E}\left[\tilde{\eta}_t\right] - \frac{L}{2}\mathbb{E}\left[\tilde{\eta}_t^2\right]\right)\|\nabla f(\theta_t)\|^2 \\
&\quad + \frac{\sigma^2 L}{2}\mathbb{E}[\tilde{\eta}_t^2]
\end{aligned}
$$

If we set $\mathbb{E}[\tilde{\eta}_t]$ small enough that

$$\mathbb{E}[\tilde{\eta}_t] - \frac{\mathbb{E}[\tilde{\eta}_t^2]L}{2} \geq \frac{1}{2}\mathbb{E}[\tilde{\eta}_t],$$

for $t \geq T_0$ which is guaranteed by the convergence of $\sum_t \mathbb{E}[\tilde{\eta}_t^2]$, then

$$\mathbb{E}\left[f(\theta_{t+1})\big|\theta_t\right] \leq f(\theta_t) - \frac{1}{2}\mathbb{E}[\tilde{\eta}_t]\,\|\nabla f(\theta_t)\|^2 + \frac{\sigma^2 L}{2}\mathbb{E}[\tilde{\eta}_t^2].$$

Now taking the full expectation

$$\mathbb{E}\left[f(\theta_{t+1})\right] \leq \mathbb{E}\left[f(\theta_t)\right] - \frac{1}{2}\mathbb{E}[\tilde{\eta}_t]\mathbb{E}\left[\|\nabla f(\theta_t)\|^2\right] + \frac{\sigma^2 L}{2}\mathbb{E}[\tilde{\eta}_t^2],$$

and summing up from $T_0$ to $T$,

$$\mathbb{E}\left[f(\theta_T)\right] \leq \mathbb{E}\left[f(\theta_{T_0})\right] - \frac{1}{2}\sum_{t=T_0}^{T-1}\mathbb{E}[\tilde{\eta}_t]\mathbb{E}\left[\|\nabla f(\theta_t)\|^2\right]$$
$$+ \frac{\sigma^2 L}{2}\sum_{t=T_0}^{T-1}\mathbb{E}[\tilde{\eta}_t^2],$$

rearranging the terms,

$$\frac{1}{2}\sum_{t=T_0}^{T-1}\mathbb{E}[\tilde{\eta}_t]\mathbb{E}\left[\|\nabla f(\theta_t)\|^2\right] \leq$$
$$\mathbb{E}\left[f(\theta_{T_0})\right] - \mathbb{E}\left[f(\theta_T)\right] + \frac{\sigma^2 L}{2}\sum_{t=T_0}^{T-1}\mathbb{E}[\tilde{\eta}_t^2].$$

Let $T \to \infty$, and notice that $\sum_t \mathbb{E}[\tilde{\eta}_t] = \infty$, $\sum_t \mathbb{E}\left[\tilde{\eta}_t^2\right] < \infty$, then

$$\sum_{t=T_0}^{\infty}\mathbb{E}[\tilde{\eta}_t]\mathbb{E}\left[\|\nabla f(\theta_t)\|^2\right] < \infty,$$

Hence $\mathbb{E}\left[\|\nabla f(\theta_t)\|^2\right] \to 0$ as $t \to \infty$. $\qquad \square$

## H. Rate-Distortion Theory and Mutual Information Bounds

We further define the *q-rate score* $R_q$ and *q-distortion score* $D_q$ as

$$R_q \triangleq \mathbb{E}_{(X,Z)\sim q}[\log q(Z|X) - \log \rho(Z)],$$
$$D_q \triangleq -\mathbb{E}_{(X,Z)\sim q}[\log \rho(X|Z)], \quad (9)$$

and similarly define *$\rho$-rate score* $R_\rho$ and *$\rho$-distortion score* $D_\rho$. Here $\{R_\rho, R_q, D_\rho, D_q\}$ are collectively referred to as the *rate-distortion scores*. We note the distortion score $D$ differs from the distortion regularizer $\mathcal{D}_t$ defined in main text, as a compromise to avoid notational clutter. The following link between rate-distortion scores and mutual information can be readily verified:

---

**Algorithm 1** Amortized FML for Latent Variable model

Learning rate schedule $\{\eta_t\}$, annealing schedule $\{\tau_t\}$, regularization strength $\lambda$
Initialize parameters $\theta, \beta, \phi$
% Optional pre-training with VAE
**for** $t = 1, 2, \cdots$ **do**
    Sample $\{x_{t,j}\}_{j=1}^m \sim \hat{p}_d(x)$, $\{z_{t,j}\}_{j=1}^m \sim q_\beta(z|x_{t,j})$,
        $\{x'_{t,j}, z'_{t,j}\}_{j=1}^m \sim p_\theta(x, z)$
    $u_{t,j} = u_\phi(x_{t,j})$,
    $I_{t,j} = \exp\{\tau_t \log p_\theta(x_{t,j}|z_{t,j}) + \log p(z_{t,j})$
              $- \log q_\beta(z_{t,j}|x_{t,j})\}$
    $J_t = \sum_j \{u_{t,j} + \exp(-u_{t,j})I_{t,j}\}$
    $D_{\rho,t} = \sum_j \log p_\theta(x_{t,j}|z_{t,j})$
    $D_{q,t} = \sum_j \log q_\beta(z'_{t,j}|x'_{t,j})$
    $D_t = D_{\rho,t} + D_{q,t}$
    $u$-update: $\phi = \phi - \eta_t \nabla_\phi J_t$
    $\psi$-update: $[\theta, \beta] = [\theta, \beta] + \eta_t \nabla_{[\theta,\beta]}\{J_t - \lambda D_t\}$
**end for**

---

**Proposition H.1** (Rate-distortion inequalities (Berger, 1971; Alemi et al., 2018))**.**

$$H(p_d) - D_q \leq I_q \leq R_q, \quad H(p_z) - D_\rho \leq I_\rho \leq R_\rho. \quad (10)$$

These bounds are tight as the proposal $q_\beta(z|x)$ approaches the true posterior $p_\alpha(z|x)$ (Lemma 5.1, Chen et al. (2016)).

## I. Algorithm of Amortized FML for Latent Variable Models

The pseudocode for latent variable FML is summarized in Algorithm 1.

## J. Connection to Langevin Gradient Flow

We remark our procedure described in Section 5 actually simulates the discrete Langevin gradient flow (Chen et al., 2018)

$$x_{t+\Delta t} \leftarrow x_t - \Delta t \nabla_x\{\log p_\theta(x_t) - \log p_{\psi^*}(x_t)\} \quad (11)$$

to solve the Fokker-Plank system

$$\partial_t p_{\theta_t} + \nabla_x \cdot (p_{\theta_t} \nabla_x \log \frac{p_{\theta_t}}{p_{\psi^*}}) = 0. \quad (12)$$

It is well known that the solution $p_{\theta_t}(x)$ of (12) converges to $p_{\psi^*}(x)$ when $t \to \infty$ under mild technical assumptions (Jordan et al., 1998).

## K. Empirical Evaluation of FML's Consistency

We experimentally verify the proposed FML is a consistent estimator, that is to say FML estimate converges to ground

truth as sample size $n$ grows to infinity.

## L. Competing Solutions

For completeness, we briefly describe competing solutions used in this study.

### L.1. Score matching (SM)

Hyvärinen (2005) proposed *score matching* (SM) to estimate an *unnormalized* density. In particular, score matching directly models the (data) score function $\nabla_x \log p_\theta(x)$, and seek to minimize the score discrepancy metric

$$
\begin{aligned}
\mathbb{F}(p_d, p_\theta) &\triangleq \tfrac{1}{2}\mathbb{E}_{X \sim p_d(x)}[\|\nabla_x \log p_d(X) \\
&\qquad -\nabla_x \log p_\theta(X)\|_2^2] \\
&= \mathbb{E}_{X \sim p_d(x)}[\Delta_x \log p_\theta(X) \\
&\qquad +\tfrac{1}{2}\|\nabla_x \log p_\theta(X)\|_2^2] + C,
\end{aligned}
\tag{13}
$$

where $C$ is a constant wrt $\theta$. Note (13) does not involve the partition function $Z(\theta)$, and other than the constant term it only depends on $p_d(x)$ through the expectation. As such, it can be easily estimated with a Monte Carlo average. A major drawback for score matching in a modern differentiable learning setting is that, the computation involves taking second-order derivatives (if the score function is directly modelled), which is costly in practice.

### L.2. Noise contrastive estimation (NCE)

Noise contrastive estimation (NCE) is a technique used to estimated the parameters for unnormalized statistical models (Gutmann & Hyvärinen, 2010; 2012), *i.e.* models with density function known up to a normalization constant. Let $p_\theta(x) = \tilde{p}_\theta(x)/Z(\theta)$ the model density function, where $\tilde{p}_\theta(x)$ is the unnormalized pdf parameterized by $\alpha$ and $Z(\theta) = \int \tilde{p}_\theta(x)\,\mathrm{d}x$ is the partition function (normalizing constant). Without loss of generality, we assume only the knowledge of $\tilde{p}_\theta(x)$ and $Z(\theta)$ is intractable. To address the intractable normalizing constant, we introduce an additional parameter $c \in \mathbb{R}$ for it, and define (unnormalized) distribution $p_\theta(x) = \tilde{p}_\theta(x)/C$, where $\theta^+ = (\theta, c)$ and $C = \exp(c)$. Note that $p_\theta(x)$ does not necessarily integrate to one. Let $p_d(x)$ be the unknown data distribution, and further introduce a contrastive distribution $q(x)$, also known as the noise distribution, which is both tractable and easy to sample from. Let $X_n = \{x_i\}_{i=1}^n$ and $Y_n = \{y_i\}_{i=1}^n$ be the respective empirical samples from data and contrastive distribution, then the contrastive objective is given by

$$
J_{\mathrm{NCE}}(\theta) = \frac{1}{2n}\sum_i \left(\log h(x_i; \theta) + \log(1 - h(y_i; \theta))\right),
\tag{14}
$$

where

$$
\begin{aligned}
h(u; \theta) &= \sigma(r(u; \theta)), & (15) \\
r(u; \theta) &= \log p_\theta(u) - \log q(u) & (16) \\
&= \log \tilde{p}_\theta(u) - \log q(u) - c,
\end{aligned}
$$

and $\sigma(t) = 1/(1 + \exp(-t))$ is the sigmoid function. This objective function is essentially the likelihood function for the class label of the mixture distribution $\frac{1}{2}[p_d + p_\beta]$, and the NCE estimate of $\theta$ is given by $\hat{\theta} = \arg\max J_{\mathrm{NCE}}(\theta)$, and we denote the corresponding model density by $\hat{p}_{\mathrm{NCE}}(x) = p_{\hat{\theta}}(x)$. NCE follows the idea of "learning by comparison", it learns the properties of the data in terms of a statistical model by discriminating the samples between data and noise. It is known that when the data distribution $p_d(x)$ is contained in the family of model distributions $\mathcal{Q} = \{p_\theta(x)\}_{\theta \in \Theta}$, then $\hat{p}_{\mathrm{NCE}}(x)$ is a consistent estimator for $p_d(x)$.

### L.3. Dynamics dual embedding (DDE)

Dynamics dual embedding (DDE) considers the primal-dual view of MLE (Dai et al., 2018). In particular, DDE exploited the following fact:

**Theorem L.1** (Theorem 1, (Dai et al., 2018)). *Let* $H(q) \triangleq -\int q(x)\log q(x)\,dx$, *we have*

$$
Z(\theta) = \max_{q \in \mathcal{P}}\{\langle q, \tilde{p}_\theta\rangle + H(q)\}
\tag{17}
$$

$$
p_\theta = \arg\max_{q \in \mathcal{P}}\{\langle q, \tilde{p}_\theta\rangle + H(q)\}
\tag{18}
$$

*where* $\mathcal{P}$ *denotes the space of distributions,* $\langle f, g\rangle \triangleq \int_\Omega f(x)g(x)\,dx$ *is the regular* $L^2$ *inner product.*

Plugging the Frenchel dual formulation of the partition $Z(\theta)$ into the likelihood estimator renders MLE into a saddle-point optimization problem:

$$
\max_{\theta \in \Theta} \mathcal{L}(\theta) \Leftrightarrow \max_{f \in \mathcal{F}} \min_{q \in \mathcal{P}} J(\theta, q)
\tag{19}
$$

where

$$
J_{\mathrm{DDE}}(\theta, q) \triangleq \mathbb{E}_{X \sim p_d}[\tilde{f}_\theta(X)] - \mathbb{E}_{X' \sim q}[\tilde{f}_\theta(X')] - H(q)
\tag{20}
$$

is the DDE objective. In the original paper, Hamiltonian flow had been used to parameterize the dual embedding distribution $q$.

### L.4. Kernel exponential family estimation (KEF)

Kernel exponential family estimation (KEF) considers the problem of nonparametric density estimation in infinite dimensional space (Sriperumbudur et al., 2017). More specifically, KEF seeks a solution of the following form

$$
p_\psi \propto \exp(-\psi_\theta(x))p_0(x),
\tag{21}
$$

where $p_0(x)$ is considered as prior regularization and $\psi(x)$ is constrained to an RKHS $\mathcal{H}_\kappa$. To match the empirical distribution $p_d$, KEF optimizes the following regularized score discrepancy:

$$\mathcal{J}_{\text{KEF}}(\theta) \triangleq F(p_d, p_\theta) + \lambda \|\psi_\theta\|_{\mathcal{H}}^2, \qquad (22)$$

where $\lambda > 0$ is the regularization strength and $\| \cdot \|_{\mathcal{H}}$ is the RKHS norm. Analytical solution can be derived with provable convergence rates.

## L.5. Stein variational gradient descent (SVGD)

Stein variational gradient descent (SVGD) (Liu & Wang, 2016) considers the problem of steepest descent in the space of probability distributions wrt KL-divergence, with descent directions constrained in certain *reproducing kernel Hilbert space* (RKHS). Formally, define the Stein operator $\mathcal{A}_p$ for $d$-dim vector function $\phi(x) \in \{\mathcal{C}^1(\Omega)\}^d$ wrt distribution $p(x)$ as

$$\mathcal{A}_p(\phi) \triangleq \phi(x) \nabla_x \log p(x)^T + \nabla_x \phi(x), \qquad (23)$$

and the *Stein discrepancy* $\mathbb{S}(q, p)$ between distribution $q$ and $p$ as

$$\mathbb{S}(q, p) = \max_{\phi \in \mathcal{F}} \{\mathbb{E}_{X \sim q}[\text{tr}(\mathcal{A}_p \phi(X))^2]\}, \qquad (24)$$

where $\mathcal{F}$ denotes some function space. Let $\kappa(x, x')$ be a semi-positive definite function known as the kernel, which defines RKHS $\mathcal{H} \triangleq \overline{\text{Span}\{\kappa(\cdot, x); x \in \Omega\}}$. Let $q_{\epsilon \phi}(x)$ be the distribution defined by the applying the following transport operator to the mass of distribution $q(x)$:

$$T_{\epsilon \phi}(x) = x + \epsilon \phi(x). \qquad (25)$$

Then it can be shown that the steepest descent direction wrt KL$(q_{\epsilon \phi} \| p)$ from the unit ball in $\mathcal{H}$ is given by

$$\phi_{q,p}^*(x) = \mathbb{E}_{X \sim q}[\mathcal{A}_p \kappa(X, \cdot)] \qquad (26)$$

with $\nabla_\epsilon \text{KL}(q_{\epsilon \phi^*} \| p)|_{\epsilon = 0} = -\mathbb{S}(q, p)$. In amortized SVGD (Wang & Liu, 2016), one optimizes the following objective to match model distribution $p_\theta$ (implicitly defined by the generator $G_\theta(Z), Z \sim p(z)$) to the unnormalized target distribution $\tilde{p}_\psi$

$$\min_\theta \mathcal{J}_{\text{SVGD}}(\theta) \triangleq \mathbb{E}_{Z \sim p(z)}[\{G_\theta(Z) - \\ \text{StopGrad}(G_\theta(Z) + \eta_t \phi_{p_\theta, \tilde{p}_\psi}^*(G_\theta(Z)))\}^2], \\ (27)$$

where $\eta_t$ denotes the learning rate.

## L.6. Generative flow (FLOW)

Generative flows (FLOW) consider modeling distribution $p_\theta$ with an generator $G_\theta(z)$ with non-degenerative tractable

Jacobian (Tabak et al., 2010). More specifically, if $G_\theta(z)$ is invertible wrt $z$, then

$$p_\theta(x_z) = p(z)|\det(\nabla_x G_\theta^{-1}(x_z))|, \qquad (28)$$

where $x_z \triangleq G_\theta(z)$. While the constraint imposed is very limiting, model flexibility can be significantly improved by stacking such simpler transformations $G_{\theta,l}(z_{l-1})$ (Rezende & Mohamed, 2015), *e.g.*,

$$\log p_\theta(x_z) = \log p(z) - \sum_{m=1}^M \log \left|\det(\nabla_{z_{l-1}} G_{\theta,l})\right|. \quad (29)$$

Different flow implementations differs in their specific choices for $G_{\theta,l}(z)$.

## L.7. Kernel density estimation (KDE)

*Kernel density estimation* (KDE) is a classical solution to the problem of nonparametric estimation of likelihood, which exploits the idea of smoothing the data with a kernel. Formally, let $\kappa(x)$ be a smoothing kernel satisfying $\int \kappa(x) \, dx = 1$, then the simplest KDE likelihood estimate is given by

$$\hat{p}_h^{\text{KDE}} = \frac{1}{nh^d} \sum \kappa \left(\frac{x - x_i}{h}\right), \qquad (30)$$

where $h > 0$ is commonly known as the bandwidth parameter. Like almost all kernel-based solution, the choice of bandwidth parameter $h$ and smooth kernel $\kappa$ are critical. Isotropic Gaussian rbf is the most popular choice for kernel, and standard practices for bandwidth selection include cross-validation based estimate and rule-of-thumb estimator $h_j = (\frac{4}{d+2})^{\frac{1}{d+4}} n^{-1/(d+4)} \sigma_j$, where $h_j$ denotes the dimension specific bandwidth and $\sigma_j$ is the standard deviation for the $j$-th dimension.

## L.8. Naive Monte-Carlo (MC)

See our discussion in Section C.

# M. Validation of the normalizing constant

In order to verify the correctness of estimated normalizing constant from FML (and NCE), we use the following estimators: ($i$) Hamiltonian annealed importance sampling (HAIS) (Sohl-Dickstein & Culpepper, 2012); and ($ii$) standard importance sampling with GMM proposal. Our implementation of HAIS is modified from the tensorflow implementation found in `https://github.com/JohnReid/HAIS`. Note sometimes the HAIS estimator will encounter numerical issues when the nonparametric estimate is not sufficiently smooth, in which cases we switch to ($ii$). We confirmed FML training yields accurate estimate of the normalizing constant. For HAIS estimator ($i$), we use $3k$ chains with 5k steps. For standard IW estimator ($ii$), we use 50 component Gaussian and draw $50k$ samples.

*Table S2.* Summary of UCI datasets

| Name | Dimension | Size |
|------|-----------|------|
| Yeast | 6 | 1358 |
| Wine-red | 8 | 1458 |
| Wine-white | 8 | 4502 |
| HTRU2 | 8 | 17898 |

## N. Toy Model Experiments

We used the KEF implementation from `https://github.com/karlnapf/kernel_exp_family`. We used our own implementation of SM, NCE and DDE. For DDE, we replace the Hamiltonian flow used in the original paper with a more expressive MAF flow. To estimate the partition function, we use a $50k$ sample MC estimator. [†] For the parameter estimation task, we rescale the results by a factor of $100$ to facilitate reading.

The exact mathematical form of the toy models we considered and the parameter specifications used are summarized below.

- *banana*: $\frac{1}{2}(((x_1 - (x_2/\kappa)^2))^2/\sigma_1^2 + ((x_2 - \mu_2))^2/\sigma_2^2)$;

- *kidney*: $\frac{1}{2} * (((\|x\| - \mu_1)/\sigma_1)^2 - \log(\exp(-.5 * ((x_1 - \mu_2)/\sigma_2)^2) + \exp(-.5 * ((x_1 + \mu_2)/\sigma_2)^2))$;

- *rings*: $\mathrm{Cat}([.25] \times 4), \mathrm{Cat}\text{-}1\,\mathcal{N}(0, \sigma_0^2), \mathrm{Cat} - i : \mathcal{N}(\|x\|; \mu_i, \sigma_i^2)$;

- *river*: $-\log(a_1(x) + a_2(x))$, where

$$a_1(x) = \exp(-.5 * ((x_2 - w_1(x, \sigma_{w,1}))/\sigma_4)^2),$$

$$a_2(x) = \begin{matrix} \exp(-.5 * ((x_2 - w_1(x, \sigma_{w,1}) \\ + w_3(x, \sigma_{w,3}, \mu_3))/\sigma_3)^2) \end{matrix}$$

- *wave*: $-\log(a_3(x) + a_2(x))$, where

$$a_3(x) = \exp(-.5 * ((x_2 - w_1(x, \sigma_{w,1}))/\sigma_3)^2),$$

$$a_2(x) = \begin{matrix} \exp(-.5 * ((x_2 - w_1(x, \sigma_{w,1}) \\ + w_2(x, \sigma_{w,3}, \mu_3))/\sigma_3)^2) \end{matrix}$$

where $w_3(x; \mu, \sigma) = 3 * sigmoid((x_1 - \mu)/\sigma), w_1(x; \sigma) = \sin(2\pi \frac{x_1}{\sigma})$. For *banana*, we use $\mu_2 = 0, \sigma_1^2 = 1, \sigma_2^2 = 2, \sigma_3 = 0.35, \sigma_4 = 0.4$ and $\kappa = 2$. For *kidney, river* and *wave* we set the parameters according to Rezende & Mohamed (2015), for *rings* we set $r_i = i$ and $\sigma_i^2 = 0.2$.

## O. UCI Data Experiments

We summarized basic info for all UCI datasets considered in this study in Table S2.

**Preprocessing.** We removed all categorical variables and normalized each dimension to zero mean and unit variance. *Entries with extreme values or missing values were removed from our analysis.*

For KDE, we use the default implementation from the *scipy* package (*scipy.stats.gaussian_kde*). Since these datasets are all high dimensional, naive uniform proposal distribution is bound to fail. In this study we first used isotropic Gaussian mixture model to fit data, then the learned Gaussian mixture model (GMM) is used as the proposal distribution for NCE and FML. We use the GMM implementation from *scikit-learn* package (*sklearn.mixture.GaussianMixture*). We specify the GMM with $50$ components and full covariance, unless this choice yields severe overfit or underfit, which is then handled on a case-by-case basis. For the flow model, we used a $4$-bijsctor MAF model with *shift_and_scale* transformations, each block has 2 hidden layers with size $256$. As standard practice, permutation layers are inserted to avoid degeneracy. Our flow model is implemented with *tensorflow* probability library package (*tensorflow_probability*). For FML and NCE, we used 3 layer feed-forward neural net to model the nonparametric potential. Each layer has $64$ hidden units.

## P. VAE Image Data Experiments

We summarized the image datasets in Table S5 and the network architectures used for respective datasets in Tables S3 and S4. As in standard VAE implementation, we used the *logit* model instead of Gaussian model at pixel level. We fixed our annealing factor to $\tau = 0.1$ in our experiments, which keeps all diagnostic statistics we used in a reasonable range (which indicates our FML is working properly, we omit details here). The results reported are from our unregularized FML implementation, regularized FML implementation show a similar trend, with improved sampling efficiency (results not shown). We use 10 latent dims for MNIST and 64 latent dims for CelebA.

## Q. GAN Image Data Experiments

To investigate how FML learning can assist the training of likelihood-free models such as GAN, we adopted the variational annealing framework (Tao et al., 2019) to regularize GAN training with FML-learned likelihood estimate. Specifically, we first encode image data using an auto-encoder and then use FML to estimate its likelihood, *e.g.*, training GAN with $\mathcal{L}_{\mathrm{VA}} = \mathcal{L}_{\mathrm{GAN}} + \lambda \log \hat{p}_\theta$, where $\mathcal{L}_{\mathrm{GAN}}$ is the standard GAN loss and $\lambda$ is the regularization parameter. We compare FML-based likelihood regularization with vanilla GAN and *denoising feature matching* (DFM) GAN, which leverages a denoising auto-encoder as score estimator (Alain & Bengio, 2014) to attain the likelihood signal. We

*Table S3.* MNIST experiment network architecture.

| Network | Architecture |
|---|---|
| Encoder | conv2d(unit=32, kernel=5, stride=2) + BN + ReLU |
| | conv2d(unit=64, kernel=5, stride=2) + BN + ReLU |
| | fc(unit=1024) + BN + ReLU |
| Decoder | fc(unit=1024) + BN + ReLU |
| | fc(unit=64*7*7) + BN + ReLU |
| | reshape to $7 \times 7 \times 64$ |
| | deconv(unit=64, kernel=5, stride=2) + BN + ReLU |
| | deconv(unit=64, kernel=1, stride=2) + BN + Sigmoid |
| $u$-net | Same as Encoder net. |

*Table S4.* CelebA experiment network architecture.

| Network | Architecture |
|---|---|
| Encoder | conv2d(unit=32, kernel=5, stride=2) + BN + ReLU |
| | conv2d(unit=64, kernel=5, stride=2) + BN + ReLU |
| | fc(unit=1024) + BN + ReLU $\Rightarrow Z$ |
| Decoder | fc(unit=1024) + BN + ReLU |
| | fc(unit=64*7*7) + BN + ReLU |
| | reshape to $7 \times 7 \times 64$ |
| | deconv(unit=64, kernel=5, stride=2) + BN + ReLU |
| | deconv(unit=64, kernel=1, stride=2) + BN + Sigmoid |
| $u$-net | Same as Encoder net. |

*Table S5.* Summary of image datasets

| Name | Dim | Train | Test |
|---|---|---|---|
| MNIST | $28 \times 28$ | 55k | 10k |
| CelebA | $64 \times 64 \times 3$ | 180k | 20k |
| Cifar10 | $32 \times 32$ | 60k | - |

*Figure S1.* Effect of variational annealing strength on MNIST

evaluated model performance on Cifar10 with IS and FID with fixed positive annealing ($\lambda = 1$, results reported in main text), and studied the effect of regularization strength $\lambda$ on MNIST using IS (see Figure S1). We used the codebase from the DFM paper for the Cifar10 experiment and implemented our own MNIST experiment.

## R. Language Data Experiments

We summarized the language datasets in Table S7 and the network architectures used for respective datasets in Table S6.

For text generation task, We use EMNLP2017 WMT News dataset and MS COCO dataset. EMNLP News dataset consist of 278686 training sentences, 10000 testing sentences, with vocabulary size 5728. MSCOCO contains 120000 and 10000 sentences for training and testing respectively, vocab-

ulary size = 27842. Our model consists of a 3-layer CNN encoder and a LSTM decoder for both datasets.

## S. Reinforcement Learning Experiments

### S.1. Soft Q-learning

Reinforcement learning seeks to maximize some reward function $r(s, a)$ wrt actions $a$ drawn from the policy distribution $\pi(a|s)$, where $s$ denotes the state. Maximal entropy

---

**Algorithm 2** FML Soft Q-learning

---

**Require:** Create replay memory $\mathcal{D} = \emptyset$; Initialize policy network parameters $\boldsymbol{\theta}$, FML network parameters $\boldsymbol{\psi}, \boldsymbol{u}$, Q network parameters $\boldsymbol{\phi}$; Assign target parameters: $\overline{\boldsymbol{\theta}} \leftarrow \boldsymbol{\theta}, \overline{\boldsymbol{\phi}} \leftarrow \boldsymbol{\phi}$. The number of samples for each distribution $M$.

1: **for** each epoch **do**
2:    **for** each $t$ **do**

3:       % Collect expereince
4:       Sample an action for $\boldsymbol{s}_t$ using $g^{\boldsymbol{\theta}}$: $\boldsymbol{a}_t \leftarrow g^{\boldsymbol{\theta}}(\boldsymbol{\xi}; \boldsymbol{s}_t)$, where $\boldsymbol{\xi} \sim \mathcal{N}(\mathbf{0}, \mathbf{I})$.
5:       Sample next state and reward from the environment: $\boldsymbol{s}_{t+1} \sim P_{\boldsymbol{s}}$ and $r_t \sim P_r$
6:       Save the new experience in the replay memory: $\mathcal{D} \leftarrow \mathcal{D} \cup \{\boldsymbol{s}_t, \boldsymbol{a}_t, r_t, \boldsymbol{s}_{t+1}\}$

7:       % Sample a minibatch from the replay memory
8:       $\{(\boldsymbol{s}_t^{(i)}, \boldsymbol{a}_t^{(i)}, r_t^{(i)}, \boldsymbol{s}_{t+1}^{(i)})\}_{i=0}^n \sim \mathcal{D}$.

9:       % Use FML to update $\psi$ and u network
10:      Sample actions for each $\boldsymbol{s}_t^{(i)}$ from the stochastic policy via
$$\boldsymbol{a}_t^{(i,j)} = f^{\boldsymbol{\phi}}(\boldsymbol{\xi}^{(i,j)}, \boldsymbol{s}_t^{(i)}; \boldsymbol{\theta}), \text{ where } \{\boldsymbol{\xi}^{(i,j)}\}_{j=0}^M \sim \mathcal{N}(\mathbf{0}, \mathbf{I})$$
11:      Follow the Algorithm.1 to estimate $\log P_{\boldsymbol{\psi}, \boldsymbol{u}}(\boldsymbol{a}_t^{(i,j)} | \boldsymbol{s}_t^{(i)})$ and update $\boldsymbol{\psi}, \boldsymbol{u}$

12:       % Update Q value network
13:      Sample $\{\boldsymbol{a}^{(i,j)}\}_{j=0}^M \sim q_{\boldsymbol{a}'}$ for each $\boldsymbol{s}_{t+1}^{(i)}$.
14:      Compute the soft Q-values $Q_{\overline{\boldsymbol{\phi}}}(\boldsymbol{a}_{t+1}, \boldsymbol{s}_{t+1})$ and calculate the loss $L_{\boldsymbol{\phi}} = ||Q_{\overline{\boldsymbol{\phi}}}(\boldsymbol{a}_{t+1}, \boldsymbol{s}_{t+1}) + r_t - Q_{\boldsymbol{\phi}}(\boldsymbol{a}_t, st)||_2$
15:      Compute gradient of Q-network and update $\boldsymbol{\phi}$

16:       % Update policy network via KL Divergence
17:      Calculate the KL Divergence, $L_{\boldsymbol{\theta}} = KL(P_{\boldsymbol{\psi}, \boldsymbol{u}}(\boldsymbol{a}_t^{(i,j)} | \boldsymbol{s}_t^{(i)}) || Q(\boldsymbol{a}_t^{(i,j)} | \boldsymbol{s}_t^{(i)}))$
18:      Compute gradient of policy network and update $\boldsymbol{\theta}$
19:    **end for**
20:    **if** epoch *mod* update_interval = 0 **then**
21:      Update target parameters: $\overline{\boldsymbol{\theta}} \leftarrow \boldsymbol{\theta}, \overline{\boldsymbol{\phi}} \leftarrow \boldsymbol{\phi}$
22:    **end if**
23: **end for**

---

*Table S6.* Summary of language datasets

| Network | Architecture |
|---|---|
| Encoder | 3-layer CNN |
| Decoder | LSTM |

*Table S7.* Summary of language datasets

| Name | Vocab | Train | Test |
|---|---|---|---|
| WMT News | 5,728 | 278k | 10k |
| MS COCO | 27,842 | 120k | 10k |

search tries to balance the exploration and exploitation wrt the policy through the following objective:

$$\mathbb{E}_{\boldsymbol{s}, \boldsymbol{a} \sim \mathcal{E}, \pi}\left[\sum_{t=0}^{\infty} \gamma^t (r(\boldsymbol{s}, \boldsymbol{a}) + \alpha H(\pi(\cdot | \boldsymbol{s})))\right], \quad (31)$$

$\alpha$ is a hyper parameter balancing the trade-off between exploitation (reward) and exploration (entropy). and $0 < \gamma < 1$ is the discounting factor. It's well known the optimal policy then would follow $\pi^*(\boldsymbol{a} | \boldsymbol{s}) \propto \exp(Q(\boldsymbol{a}, \boldsymbol{s})/\alpha)$, where $Q(\boldsymbol{a}, \boldsymbol{s})$ is known as the Q-function (Sutton & Barto, 2018). Soft-Q learning (Haarnoja et al., 2017) leverages the current policy $\pi_t$ to interact with the environment $\mathcal{E}$ to update Q-function estimate $\hat{Q}(\boldsymbol{a}, \boldsymbol{s})$, and then train the policy towards the optimal distribution defined by $\hat{Q}(\boldsymbol{a}, \boldsymbol{s})$.

**SVGD-SQL** In Soft Q-Learning (Haarnoja et al., 2017), the policy network is trained amortised in two steps: (1), draw action from the policy network and use these actions as the initial points for SVGD update. (2), use the $\ell_2$ distance bewteen the origin samples and the updated ones to calculate the gradient of the policy network. The first step suffers from a risk that the updated samples are out of the action space. Constraints should be added to prevent from this, leading to the unexpected errors.

**S.2. Experimental setup**

Detailed architectures and parameter setting used in our experiments are summarized in Tables S9 to S9. We used the notation "X–H–H–Z" to denote a network with X as the in-

put size, H as the number of hidden units and Z is the output size, and the notation $+$ denotes concatenation. Rectified linear units (*ReLU*) are used as the activation function for the hidden layers in all our experiments. Hyper-tangent (*tanh*) activation is applied to the policy network's output. $\mathcal{N}$ denotes standard Gaussian noise with the same dimension as the action space.

*Table S8.* Environment hyper-parameters for SQL experiments.

| Environment | Action Space Dim | Reward Scale | Replay Pool Size |
|---|---|---|---|
| Swimmer (rllab) | 2 | 100 | $10^6$ |
| Hopper-v1 | 3 | 1 | $10^6$ |
| Walker2d-v1 | 6 | 3 | $10^6$ |
| Reacher-v1 | 2 | 10 | $10^6$ |

*Table S9.* Neural architectures used for SQL experiments.

| Network | Architecture |
|---|---|
| Policy-Network | $|\mathcal{S} + \mathcal{N}|$–128–128–$|\mathcal{A}|$ |
| Q-Network | $|\mathcal{S} + \mathcal{A}|$–128–128–1 |
| $\Phi$-Network | $|\mathcal{S} + \mathcal{A}|$–128–256–1 |
| $b$-Network | $|\mathcal{S}|$–128–256–1 |

*Table S10.* Training hyper-parameters for SQL experiments.

| Hyper-parameters | Values |
|---|---|
| Learning rate for policy-net | $3 \times 10^{-4}$ |
| Learning rate for Q-net | $3 \times 10^{-4}$ |
| Batch-size | 128 |
| SVGD particle size | 32 |