[Reviews · NeurIPS 2019]

Reviewer 1



My only concern is regarding the training stability. The computational difficulty arises in calculating the integration of $I(x; \phi_\theta)$ in an efficient and reliable way. The authors propose to use importance sampling for approximation. However, it is widely known that importance weighting suffers from extremely high variance in cases where the proposal and target distributions are not a good match, especially for high dimensional cases. \E[p/q] and \var[p/q] could even be infinite in lots of scenarios. Also, it’s unclear to me about how to select an approximate $q$ in a principal way, since a good setting might reasonably be expected to be dependent on the learning task that is being considered. According to the pseudo-code in Algorithm 1, it appears the number of MC samples at each iteration is just 1. Does the algorithm converge in general, how to choose q? Could the authors offer more insights?

Reviewer 2



(This reads like a critical review, which may appear to conflict with my rating, because I have concerns about the validity and/or practical utility of the claimed contributions, so I was initially leaning towards recommending rejection. However, after I read the supplementary material, I realized there might be an important contribution that the authors didn't emphasize and that I had missed. I then re-read the paper and decided to substantially upgrade my rating. I would strongly urge the authors to re-write the paper for the camera-ready to focus on this contribution.) Summary of main contribution (in my view): It is easy to obtain a Monte Carlo estimate of the partition function - while such an estimate is unbiased, the log of the estimate is an underestimate of log-partition function. This means that an estimate for the log-likelihood constructed using this estimate *overestimates* the log-likelihood, which causes many issues in practice because it is not good to think the model is doing better than it actually is. Prior work (notably, RAISE [a]) has developed a way of overestimating the log-partition function and therefore underestimating the log-likelihood. But to my knowledge, there does not exist a way of estimating the log-partition function and the log-likelihood in an unbiased fashion. [a] Burda at al, "Accurate and conservative estimates of MRF log-likelihood using reverse annealing", AISTATS 2015 The proposed method provides such a way. It works by applying a simple transformation, namely the Fenchel conjugate of -log(t). On the surface, this looks like a step backwards, since a function that can be computed analytically is replaced with an optimization problem that is to be solved numerically. However, the benefit is that the log doesn't appear in the objective function, which makes it possible to estimate in an unbiased fashion using a Monte Carlo estimate. I would strongly recommend the authors to re-write the paper and focus on this contribution. Currently, the bias of the naive Monte Carlo estimator is only discussed in the supplementary material (which was also discussed in [a]); it should instead be moved to the main paper. Moreover, empirical results related to this should be highlighted, e.g.: Fig. 1. I would also recommend plotting the log-likelihood rather than the likelihood in Fig. 1, since the log-likelihood is what the proposed method is able to estimate in an unbiased fashion. It would also be useful to show some results on higher-dimensional data and compare to other approaches, including [a]. I wouldn't be surprised if the proposed method doesn't perform well in high dimensions because of the increased variance of Monte Carlo estimates, but even in that case, I believe this contribution is important because even in low dimensions, there isn't a way to estimate the log-likelihood in an unbiased manner. Critique of the paper as it was originally presented: My biggest issue with this paper is that it claims to sidestep the estimation of the partition function, when it in fact does not. For example, on L138-139, the paper claims that "unlike existing solutions, FML targets the exact likelihood without using finite-sample estimator for the partition function". While technically true, the proposed method uses a finite-sample estimator for the reciprocal of likelihood, which carries the same issues as a finite-sample estimator for the partition function. The difficulty of learning probabilistic models in most cases comes from computing/estimating the partition function, and the alternative formulation proposed in this paper does not make this any easier. A naive Monte Carlo estimate of the partition function suffers from high variance in high dimensions. While the paper uses importance sampling to estimate the likelihood, the main challenge to controlling the variance is to choose a good proposal distribution that is both tractable and assigns high density to regions in the data space where the likelihood is high. Yet, the minimax formulation does not sidestep this issue, and the question of how to choose a proposal distribution is left largely unaddressed by this paper. So, it is unclear what is gained by having the minimax formulation over the plain old MLE formulation. Certainly, one fo the claimed advantages doesn't make sense. On L131, the paper highlights that under the proposed formulation, "we have isolated learning the partition function and the model parameters". Why is this useful? The challenge comes from computing the partition function, and so the goal of various approaches is to learn the model parameters *without* needing to compute the partition function. If we had an oracle that can compute the intractable partition function, then naive gradient ascent on the log-likelihood would work fine. If we don't have such an oracle, we can try to compute the partition function in each iteration of gradient ascent on the parameters, and so the partition function computation and learning are also done separately in this case. This is akin to what the proposed method does, which estimates the partition function in the inner loop. Also, on L127, it is unclear why b_\theta can be set to a particular value. Because u_\theta is decomposed into the sum of \psi_\theta and b_\theta and \psi_theta is given, it appears that b_\theta must be a free parameter (otherwise, there would be nothing to learn). If b_\theta is indeed a free parameter, then what is driving b_\theta to be equal to the log-partition function? Finally, it is unclear if the proposed formulation is actually a minimax formulation (despite the title and the name of the proposed method). From eqn. 3, it seems that one can move the minus sign before the min inwards to yield a pure maximization formulation (unless there's a typo somewhere else that I didn't catch). Other Issues: L13: The footnote is unnecessary - this is common knowledge. L32: "any finite-sample Monte Carlo estimate" - "any" is too strong; I would suggest changing it to "existing". Certainly the proposed method uses a Monte Carlo estimate and is unbiased. L37: One distinguishing feature of CD that is different from prior methods is the particular way of initializing the Markov Chain, which should be noted. L44: What is the difference between "poor scalability" and "computational efficiency issues"? L44: Also, it should be clarified which version of CD results in biased estimation. Is it the version where the chain is run for a finite number of steps or an infinite number of steps? If it's the latter, it should be explained why CD is biased. L46-50: It is important to separate models without explicit unnormalized likelihoods and particular training techniques, like adversarial learning. Models without explicit unnormalized likelihoods are typically referred to as implicit models, see e.g.: [b], and the generator in GANs can be viewed as an example of such a model. However, there are also other techniques for training such models, like GMMN [c,d] and IMLE [e]. [b] Mohamed & Lakshminarayanan, "Learning in implicit generative models", Arxiv 2016 [c] Li et al., "Generative moment matching networks", ICML 2015 [d] Dziugaite et al., "Training generative neural networks via maximum mean discrepancy optimization", Arxiv 2015 [e] Li & Malik, "Implicit maximum likelihood estimation", Arxiv 2018 Alg. 1: \hat{p}_d is not defined - I assume it means the empirical distribution. L164: For Robbins-Monro to hold, you need strict convexity as well. L189: There is no need to refer to GANs here, because you are just defining a latent variable model with a Gaussian prior and a Gaussian condition likelihood whose mean is given by some function. This is the same as the decoder in a VAE, and there is no reason to say this model can be viewed as an approximation to the generator in GANs as the variance tends to zero. Also, the notation seems to be specific to the scalar case with 1D observations and could be generalized. Finally, there was a paper other than [23] that independently discovered GANs [f], and should be cited for completeness. [f] Gutmann et al., "Likelihood-free inference via classification", Arxiv 2014

Reviewer 3



This paper presents Fenchel Mini-Max Learning formulation for density estimation, with the goal of being able to do inference, parameter estimation, sampling and likelihood evaluation in a tractable manner. Major comments: 1. I think equation 2 is incorrect and the negative sign on the left side should be removed; if in the conjugacy relationship in line 114, we use t = 1/p(x), then we reach (2), without the negative sign on the left side. This propagates to equation (3) too, and might affect the implementation. please double check. 2. The formulation relies on the importance sampling estimation of the normalization constant (I(x, \psi)), i.e., you are basically using Monte Carlo estimation of 1/p(x). This is in contrast with what is states in lines 138-140. Also, the quality of proposal distribution then can significantly affect the quality of solutions by FML. I think there exists a lack of discussion on this matter in the current manuscript. 3. In the gradient analysis part, authors resort to Monte Carlo estimate for the inverse likelihood (equation (4)). What is the benefit of this over just getting a Monte Carlo estimate of the likelihood itself? 4. The formulation of FML mostly relies on the Fenchel conjugacy for log(t) function. Can this framework be applied for a general convex function f?

[Author Response · NeurIPS 2019]

We thank the reviewers for their insightful and constructive comments.

*Reviewer #1 and shared comments.*

• ***(Shared by R2, R3) Stability of importance sampling, discussion & analysis on choice of proposal*** $q(x)$***.*** This is
a challenge shared by almost all non-parametric density estimation models (e.g., NCE, DDE). Experimentally, our FML
outperforms its counterparts which also involve importance sampling estimates. To keep the variance in check, a general
guiding principle for choosing a good $q(x)$ is to make it as close to target $p(x)$ as possible, which is expected to yield
small var[p/q]. To this end, for explicit FML we have proposed to use a pre-trained tractable sampler $q(x)$ modeled
with generative flows (SM L.6, likelihood maximized wrt empirical data; other models like GMM are also applicable).
For latent FML we maximize the mutual information (Sec 3.3). We have revised the paper to expand the theoretical
discussions and elaborate implementation details on the choice of $q(x)$. Empirical comparisons with different proposals
are also added for sensitivity analysis and show the gains with a good proposal, with both simple and complex datasets.

• ***MC samples & convergence.*** We remind the reviewer that one of the key features of our FML framework is that we
replace the direct estimation of normalizing constant (typically requires multiple MC samples) with an optimization
procedure, such that under the SGD setup 1 MC sample suffices. For convergence guarantees, we have proved with FML
model parameters converge to the correct answer under both convex setting (Col 2.3) and more general non-convex
setting (SM Thm G.2). Other competing MC-based solutions generally cannot guarantee this under finite sample.

*Reviewer #2.* We thank the reviewer for this very comprehensive review, which we really appreciate.

• ***Highlighting the contribution of unbiased estimation of likelihood.*** We agree this point needs to be reinforced. It
is the key motivation of this study and we have rewritten relevant sections in the paper to reflect the reviewer's inputs.
We have also added the discussion of the biased-estimation issue to the main part and updated the figs as suggested.

• ***Finite sample estimate of the partition.*** Our FML treats the partition function as a learnable parameter that is
updated with *finite sample evaluations* of the inverse likelihood, so that the objective does not involve a $\log$ transform.
Technically it is not a (direct) finite sample estimator. This differs from a direct $\log$ (finite sample estimate) adopted by
competing solutions, which lead to biased estimation/gradient of likelihood, a key challenge that FML addressed. We
agree FML itself cannot sidestep the challenge of choosing efficient sampling schemes for the evaluation of the inverse
likelihood integral (i.e., choice of proposal $q(x)$ used in SGD), which is discussed in detail in our reply to R1 above.

• ***What's gained by the minimax game over plain MLE.*** In our FML the log-partition is modeled as a learnable
parameter, and our theory guarantees convergence to the correct answer as long as the log-partition is estimated with
bounded error. The major gain of minimax FML is unbiased estimation for *unnormalized* statistical models and latent
variable models, where the exact likelihood is intractable and existing solutions typically settle for bounds.

• ***L127 Is*** $b_\theta$ ***fixed or learned.*** This is a misunderstanding that will be made more clear in our revision. $\log$-partition
estimate $b_\theta$ is a free-parameter to be learned, and $b_\theta$ minimizes the objective iff it equals to the true log-partition.

• ***Why called a minimax formulation.*** We agree the explicit FML (Eq 3) can be understood as a min-min game, but
since the latent FML formulation (Eq 7) is a strict min-max game, calling it a minimax game is more consistent.

• ***Response to improvement suggestions.*** We have rewritten relevant sections to highlight that our FML provides
an unbiased estimate of the log-likelihood using the Fenchel mini-max setup as a key contribution, addressing a
long-standing challenge in statistical estimation. We will remove, rephrase or clarify the claims the reviewer found
inaccurate/confusing/unjustified. While current submission already includes experiments on high-dimensional complex
data (e.g., image, language) with the latent variable FML, we will report more results with explicit FML in our revision.

• ***Misc issues.*** We thank the reviewer for mentioning additional relevant literature ([a-f]), which have been updated to
our draft with the suggested discussions. The manuscript has been revised to clarify CD and correct tech conditions
used in our theory. Edits are also made to incorporated all other suggestions, which further improved our presentation.

*Reviewer #3.*

• ***Derivation of Eq 2.*** Our math as presented is correct; the reviewer must have missed a minus sign somewhere.
To derive Eq 2, let $t = \frac{1}{p(x)}$ and we have $-\log p(x) = -(-\log t) = -(\max_u\{-u - \exp(-u)t + 1\}) = \min_u\{u +$
$\exp(-u)t - 1\}$. Our SM includes more on this equation, see our code there for implementation details.

• ***What's the benefit of using inverse likelihood MC evaluation over direct likelihood estimate.*** To understand the
benefits we need to clarify how we estimate the likelihood (Eq 3) with how we compute the gradient for model updates
(Eqs 4-6). FML likelihood is estimated through optimization (Eq 3, min step), and later used to adjust for the scaling
of model parameter gradient (Eq 6) computed from MC inverse likelihood evaluations (Eq 4, unbiased). As a result,
FML guarantees the model parameters $\theta$ will converge to the right answer even with less accurate likelihood estimate
(bounded error, Col 2.3, SM Thm G.2). On the other hand, computing the gradient directly with a direct MC likelihood
estimate introduces bias when updating model parameters (SM Sec C), and there is no guarantee of convergence with
finite samples.

• ***Can it be generalized.*** The Fenchel conjugacy technique is applicable for other convex functions, with which more
general likelihood evidence scores can be defined (ref [54]). However, (a) such criteria are less popular in practice; (b)
Fenchel conjugacy does not necessarily have a closed form for an arbitrary convex function; and (c) unlike $\log(t)$ the
unbiased estimation cannot be guaranteed in general. Further investigation is warranted for future study.

[Meta-Review · NeurIPS 2019]

The reviewers have discussed this paper and feel that the author response addressed the majority of their concerns. I recommend that the authors take these comments to heart in editing the paper for the camera ready version of the paper, as this will help the community appreciate the new ideas within and discuss it effectively.